# Targeting ACE2 with a camelid antibody inhibits SARS-CoV-2 binding and has protective effects in vivo

Simon Blachier [1], Marie-Christine Vaney [2], Laurine Conquet[3,4,12], Isabelle Staropoli[5,12], Ignacio Fernández[2], Emilie Giraud [6], Atousa Arbabian[2], Vincent Michel [7], Fruzsina Szilagyi [1], Salomé Guez[6], Alix Boucharlat[6], Jeanne Chiaravalli[6], Jaouen Tran-Rajau[6], Evelyne Dufour[8], Ahmed Haouz[9], Stéphane Petres[8], Delphine Planas [5,10], Xavier Montagutelli [3], Fabrice Agou [6], Pierre Lafaye [11], Gabriel Ayme [11,13], Olivier Schwartz [5,10,13], Felix A. Rey [2,13], Jost Enninga [1,13] & Anne Brelot [1] ✉

The continuous emergence of antibody-escape variants of SARS-CoV-2 demands the identification of alternative methods of protection against infection that do not directly target viral proteins. Here, we generated heavy-chain-only antibody (VHHs) from an alpaca immunized with the human angiotensin-converting enzyme 2 (hACE2), the major entry receptor for SARS-CoV-2. The VHHs bind hACE2 without affecting its enzymatic activity, and two of them (B07 and B09) inhibit all SARS-CoV-2 isolates tested (Delta, BA.1, BQ1.1, XBB.1.5, XBB.1.16.1, EG.5.1.3, BA.2.86.1). Their X-ray structure in complex with hACE2 show that their epitope overlaps with the footprint of the receptor binding domain (RBD) of the SARS-CoV-2 spike on hACE2. A dimeric B07-Fc fusion construct avidly binds hACE2 with an apparent dissociation constant of 0.1 nM and inhibits in vitro infection of previously tested variants and, of JN.1.1 and KP.3.3 variants, with an IC50 ~ 1 nM. In vivo experiments using K18-hACE2 mice show that intranasal prophylactic administration of B07-Fc confer a dose-dependent protection against SARS-CoV-2 D614G and Omicron variants. These VHHs targeting hACE2 represent potential broad-spectrum therapeutic candidates against potential new emerging coronaviruses using hACE2 as a receptor.

Severe Acute Respiratory Syndrome Coronavirus 2 (SARS-CoV-2) is the causative agent of COVID-19 (coronavirus disease 2019). So far, it has infected more than 770 million people with more than 7 million deaths worldwide, even though these numbers are likely underestimated (World Health Organization – WHO, 2024). The clinical use of COVID-19 vaccines and neutralizing antibodies, both targeting the viral spike (S) protein, has helped control the pandemic[1-3]. However, the emergence of new variants represents a continuing threat of resurgence. The latest Omicron subvariants (KP.3, XEC) contain more than 60 mutations in S compared to the wild type isolate[4], resulting in antigenic escape[5,6], such that previous neutralizing antibodies became poorly protective[7,8]. Constantly updating vaccines and therapeutic antibodies is therefore critical[9,10]. Two strategies have been proposed to combat antigenic escape: monoclonal antibody (mAb) cocktails or bispecific antibodies[11]. While both approaches can reduce viral escape, neither is currently in clinical use due to manufacturing constraints to produce these antibodies in time. Other strategies are therefore needed to complete the arsenal against COVID-19.

SARS-CoV-2 infects cells by binding its S protein to the human angiotensin-converting enzyme 2 (hACE2), an interaction that triggers fusion of viral and cell membranes[12]. hACE2, a single trans-membrane glycoprotein expressed in almost all tissues[13], is responsible for the regulation of the Renin-Angiotensin-System[14]. Besides its cellular form, hACE2 also exists as a soluble enzymatically active form circulating in the plasma[15]. Soluble ACE2 may indirectly be involved in SARS-CoV-2 entry[16]. Using soluble derivatives of ACE2 as a "decoy" to trap the S protein is an approach being studied to avoid viral escape[17–19].

Here, we developed a strategy for the direct targeting of trans-membrane hACE2 instead of S. Because of the major role of hACE2 in regulation of blood pressure and in vascular, renal, and myocardial physiology, the challenge was to develop compounds, targeting the S-binding surface on hACE2 without affecting its catalytic activity. The enzyme's active site and its S-binding surface are located in distinct regions of hACE2, suggesting that this approach is feasible[20].

Monoclonal antibodies targeting hACE2 proved their efficacy in inhibiting SARS-CoV-2 infection of multiple variants[21–25]. An alternative to mAbs for clinical application is the use of camelid single chain antibodies, in particular the variable region of their heavy-chain (also called VHHs or nanobodies). These VHHs, 10 times smaller than "conventional" antibodies, retain full antigen-binding capacity, have a propensity to bind small epitopes, and are easily engineered[26]. For example, VHHs are amenable to covalent linkage with each other or with the Fc domain, thereby increasing their valency and potency, and improving their pharmacodynamic profile in vivo[27–29]. These features enable the development of inhalable preventive formulations, ensuring direct delivery to the lungs, thereby conferring significant advantages to the treatment of respiratory diseases. Such treatments have been used successfully for other diseases. For example, a VHH-based therapy was approved in the U.S. and Europe in 2019 for the treatment of acquired thrombotic thrombocytopenic purpura[30].

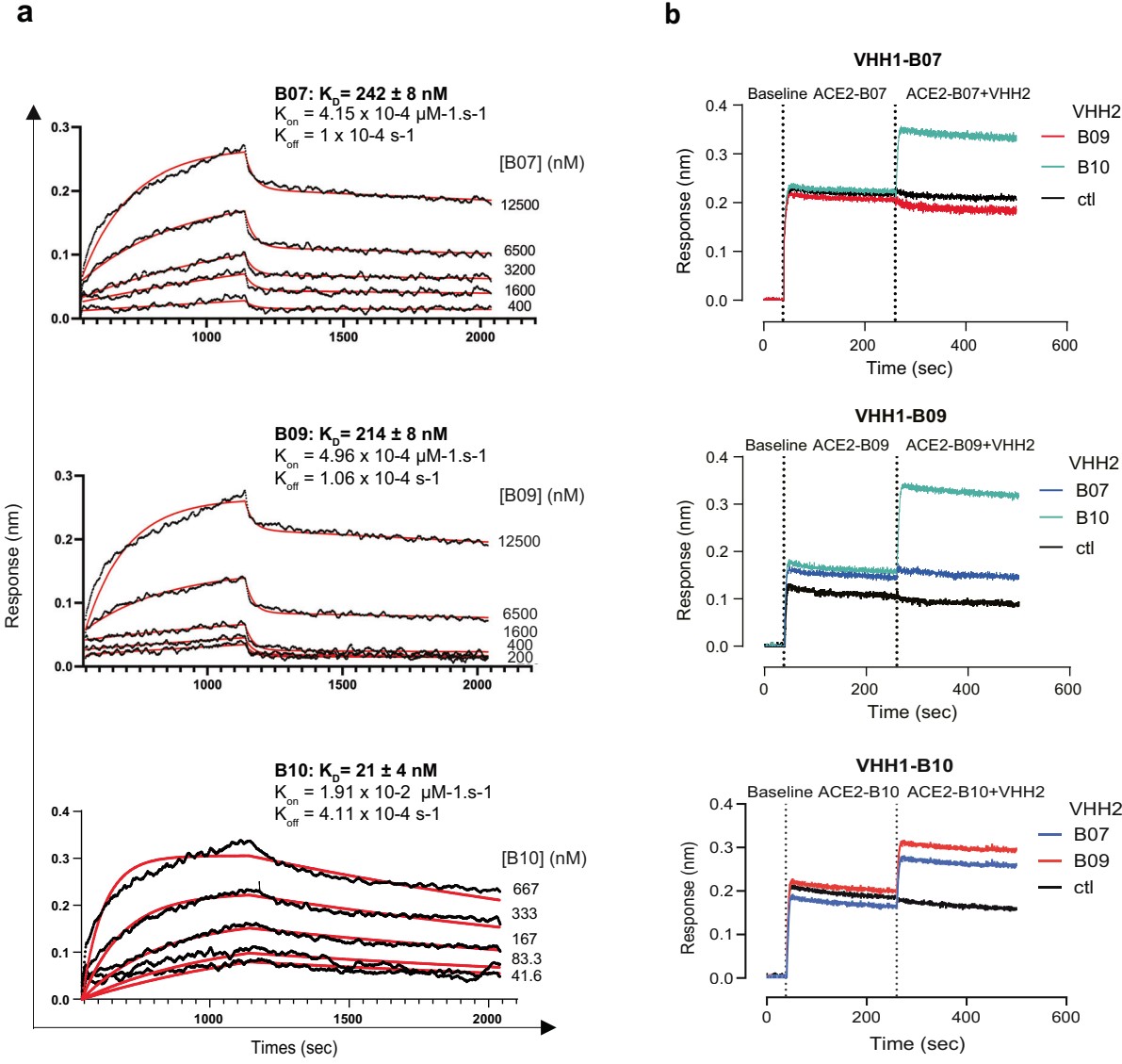

**Fig. 1 | VHHs binding on soluble human ACE2 and epitope mapping. a** Kinetic analysis of B07, B09, and B10 VHHs on hACE2-Fc by BioLayer Interferometry (BLI) using various concentrations of VHHs (as indicated on the figure). AHC biosensors were used to immobilize hACE2-Fc. $K_D$ of each VHHs is indicated. One representative experiment out of two. **b** Epitope mapping. sACE2-Fc (5 µg/mL) was immobilized onto the AHC biosensors. After a baseline step, a first VHH was applied (VHH1; 5 µg/mL). The sensor was dipped in a mixture of VHH1 and the competitor VHH2 at the same concentration (5 µg/mL). An anti-IgM specific VHH was used as control (ctl). (Top) Immobilized sACE2 pre-bound to B07 (VHH1) and VHH2 being B09, B10, or IgM (ctl). (Middle) Immobilized sACE2 pre-bound to B09 (VHH1) and VHH2 being B07, B10, or IgM (ctl). (Bottom) Immobilized sACE2 pre-bound to B10 (VHH1) and VHH2 being B07, B09, or IgM (ctl). Source Data are provided as a Source Data file.

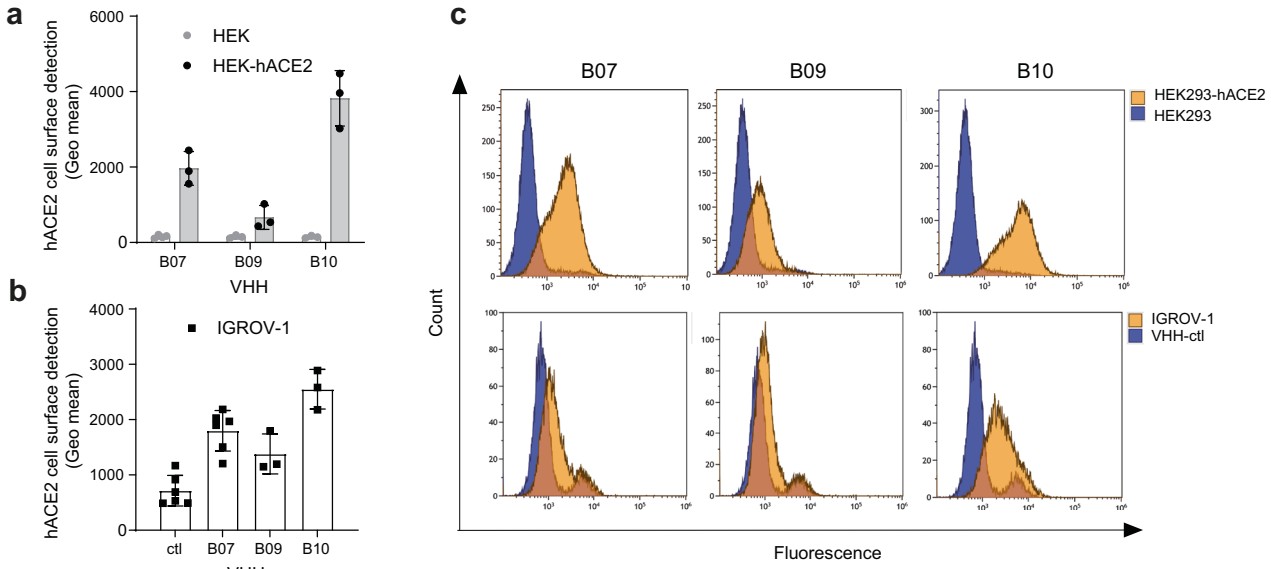

**Fig. 2 | VHHs binding on cells expressing hACE2.** Cells were incubated with the different VHHs (10 μg/mL), stained with an anti-myc antibody and a AF488-conjugated anti-mouse antibody, before being analyzed by flow cytometry. **a** B07, B09, B10 binding efficacy on HEK293-hACE2-expressing cells. Data are means ± SD of three independent experiments. Parental HEK293 cells were used as control. **b** B07, B09, B10 binding efficacy on cells expressing endogenous hACE2 (IGROV-1).

Anti-IgE VHH was used as a control (ctl). Data are means ± SD, n = 6 (ctl, B07) or 3 (B09, B10) independent experiments. **c** Fluorescence diagram overlays: B07, B09, B10 efficacy on cells expressing exogenous (HEK293-hACE2) or endogenous hACE2 (IGROV-1). Background (blue) corresponds to the fluorescence intensity obtained on parental cells (HEK293) or using a VHH control (anti-IgE VHH) (IGROV-1). Source Data are provided as a Source Data file.

In this study, we produced monomeric VHHs and a dimeric VHH fused to a human IgG1 Fc-fragment that (1) specifically bind soluble hACE2, (2) recognize hACE2 at the surface of different cell types, (3) do not alter the enzymatic activity of hACE2 neither in vitro nor *in cellula*, and (4) efficiently inhibit multiple variants of SARS-CoV-2 as tested in vitro (Delta, BA.1, BQ.1.1, XBB.1.5, XBB.1.16.1, EG.5.1.3; BA.2.86.1, JN.1.1, KP.3.3). These VHHs engage their 3 complementarity-determining regions (CDRs) to compete with S binding by targeting residues primarily located in helix α1 of hACE2. Moreover, prophylactic intranasal administration of the dimeric VHH-Fc protected K18-hACE2 mice and hamsters against SARS-CoV-2.

## Results

### Selection of VHHs targeting human ACE2
We immunized an alpaca with the soluble catalytic ectodomain of human ACE2 (sACE2) and we selected three specific VHHs by phage display, designated B07, B09, and B10. Sequence analysis revealed that B07 and B09 originate from the same gene (*IGHV3S53*) and have the same junction region (Supplementary Fig. 1), indicating they derived from the same lineage. They underwent different somatic hypermutations (15.5% and 12.9% amino acid difference with the germline for B07 and B09, respectively) but differ by only six amino acid mutations. In contrast, B10 is encoded by a different gene (*IGHC3-3*) and has a totally different junction region, indicating that it was derived independently from the two other VHHs (Supplementary Fig. 1).

### VHHs B07, B09, and B10 interact with soluble ACE2
The different VHHs were expressed in fusion with His and c-myc tags at the C terminal end. We characterized the interaction between VHH and soluble hACE2 by determining association ($k_{on}$) and dissociation ($k_{off}$) rate constants using biolayer interferometry (BLI) technology (Octet HTX, Sartorius)[31] (Fig. 1a). We immobilized hACE2-Fc on AHC biosensors and incubated them with different concentrations of VHHs. From these data, we deduced a $K_D$ value of 242 nM, 214 nM, and 21 nM for B07, B09, and B10, respectively (Fig. 1a), indicating that these VHHs bind sACE2 with moderate (B07, B09) and high (B10) affinities. We

confirmed these findings using VHH-His immobilized on Ni-NTA biosensors and different concentrations of sACE2 injected as analytes ($K_D$ value of 1141 nM, 1961 nM, and 22 nM for B07, B09, B10, respectively) (Supplementary Fig. 2a).

To obtain insights into the epitopes recognized by the three VHHs, we carried out a binding competition assay using the BLI system (Fig. 1b, Supplementary Fig. 2b). B09 or B07 did not bind the immobilized hACE2 pre-bound to B07 (VHH1-B07) or B09 (VHH1-B09) respectively, suggesting that these two VHHs recognize the same, or an overlapping epitope on hACE2 (Fig. 1b). On the contrary, B10 bound VHH1-B07 or VHH1-B09, suggesting that this VHH recognizes a different epitope than B07 and B09 (Fig. 1b). In agreement with this finding, the immobilized hACE2 pre-bound to B10 (VHH1-B10) was still able to bind B07 or B09 (Fig. 1b).

### VHHs B07, B09, and B10 bind ACE2 at the surface of cells
Next, we tested the ability of the three VHHs to bind ACE2 present at the surface of different cell lines, which express exogenous (HEK293-hACE2, A549-hACE2) or endogenous human and non-human ACE2 (IGROV-1, Vero E6) (Fig. 2, Supplementary Fig. 3). We estimated ACE2 cell surface expression for each cell line by flow cytometry using a commercial ACE2 mAb (AF933) or assessing spike binding (Supplementary Fig. 3, a-c). Then, we incubated 10 μg/mL (~667 nM) of each VHH with the different cell lines or with parental cells, and we used an unrelated myc-tagged VHH as negative control (ctl). The myc-tagged VHHs were detected bound to the cells' surface by flow cytometry after staining with an anti-myc antibody and a fluorescent secondary antibody (Fig. 2, Supplementary Fig. 3d, e). The three VHHs were specifically detected at the surface of all cell lines, revealing their interaction with trans-membrane ACE2. B10 showed the highest staining in accordance with its high affinity for sACE2 defined by BLI ($K_D$ 21 nM) (Fig. 1a).

### VHHs B07, B09, and B10 do not impact the enzymatic activity of hACE2
We determined whether B07, B09, and B10 VHHs affected hACE2 function (Fig. 3). hACE2 catalyzes the cleavage of small circulating

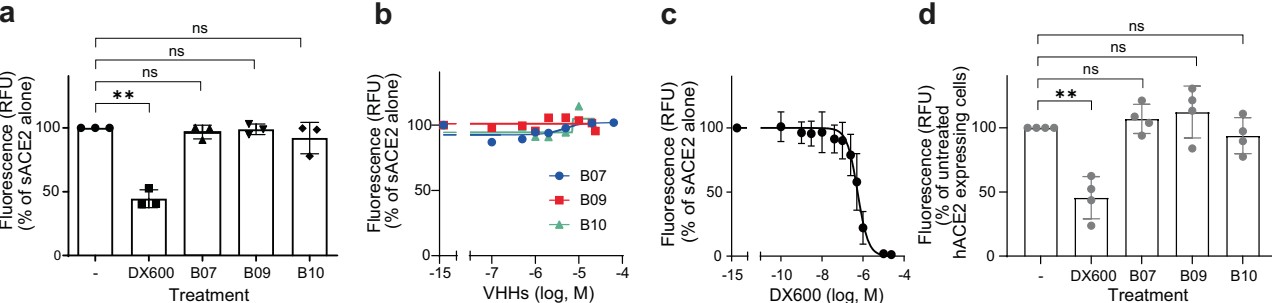

**Fig. 3 | Enzymatic activity of soluble and membrane hACE2.** hACE2 activity in the presence of VHHs B07, B09, B10 or absence (-) was assayed using the SensoLyte 390 ACE2 Activity Assay Kit, which measures fluorogenic peptide cleavage. **a** Soluble hACE2 (sACE2) activity in the presence of 10 μM VHHs. An ACE2 inhibitor, provided with the kit, was used as control (DX600, 1 μM). Data are means ± SD of three independent experiments. Paired T test two-sided: ns, nonsignificant; ** $p = 0.0053$. **b** VHHs B07, B09, B10 were used at different doses (one representative experiment out of three performed in duplicates. Mean values of the duplicate).

The mock control (buffer) was arbitrarily positioned at $10^{-14}$. **c** The ACE2 inhibitor DX600 was used at different doses. Data are means ± SD of seven independent experiments. The mock control (buffer) was arbitrarily positioned at $10^{-14}$. **d** Cell surface hACE2 activity in the presence of saturating concentration of VHHs (> 20 μM) or 1 μM DX600. Data are means ± SD of four independent experiments. Paired *T* test two-sided: ns, nonsignificant; ** $p = 0.0070$. Source Data are provided as a Source Data file.

peptides, thus playing a central role in the control of the renin-angiotensin system[14]. We assayed the impact of 10 μM of each VHH (Fig. 3a), or different doses (Fig. 3b), on the ability of sACE2 to cleave a fluorogenic peptide substrate, Mc-Ala/Dnp where Mc-Ala fluorescence is quenched by Dnp (SensoLyte assay). hACE2 enzymatic cleavage separates the two fragments, allowing recovery of Mc-Ala fluorescence, which is monitored at excitation/emission wavelength = 330/390 nm. We found no effect of the three VHHs on peptide cleavage (Fig. 3a, b), even at 40 μM of VHHs (Fig. 3b), contrary to the specific ACE2 inhibitor DX600, used as positive control (IC50 = 0.54 μM) (Fig. 3c). We confirmed this result *in cellula* by following peptide cleavage using HEK293-hACE2 live cells instead of soluble hACE2 (Fig. 3d). This result suggested that the interaction of B07, B09, and B10 VHHs with soluble hACE2, or cell surface hACE2, did not interfere with its enzymatic activity.

## VHHs B07 and B09 inhibit infection of multiple SARS-CoV-2 variants by competing with S binding to hACE2

We examined whether the selected VHHs may inhibit the ability of SARS-CoV-2 infected cells to form syncytia using the S-Fuse assay[32]. We pre-incubated S-Fuse cells (U2OS-hACE2 GFP1-10 and GFP11) with serial dilutions of VHHs, exposed them to different SARS-CoV-2 variants (Delta, BA.1, BQ.1.1, XBB.1.5, XBB.1.16.1, EG.5.1.3; BA.2.86.1) and quantified syncytia formation 18 h after infection.

B07 and B09, which are closely related (Supplementary Fig. 1), inhibited syncytia formation induced with the 7 SARS-CoV-2 variants tested with similar IC50 (from 1.45 to 30 μg/mL corresponding to 95.1–1937 nM) (Fig. 4a, Supplementary Table 1a, b). In contrast, B10 which has the highest affinity for hACE2 (Fig. 1a), displayed no inhibition of syncytia formation induced by the different variants tested (Fig. 4a). This result is in line with B10 targeting a different epitope on hACE2 than B07 and B09 (Fig. 1b), most likely not involved in S recognition.

To assess whether B07 and B09 inhibited SARS-CoV-2 infection by inhibiting binding of the S protein, we performed competition experiments in which binding of soluble S (ancestral spike) on HEK293-hACE2 cells was measured in the presence of increasing concentration of VHHs B07 and B09. B10 was used as control. S binding was assessed after immuno-staining with an anti-S antibody and flow cytometry (Fig. 4b). Pre-incubation of HEK293-hACE2 with VHHs B07 and B09 prevented soluble S binding, while B10 had no effect (Fig. 4b). This suggested that B07 and B09 inhibited SARS-CoV-2 infection by inhibiting hACE2-spike interaction.

We then performed competition experiments with B07 or B09 using increasing concentrations of S from 5−10 μg/mL (Fig. 4c). The relationship between the observed IC50 value and tracer concentration (here S) in displacement experiments indicates whether binding inhibition is competitive (i.e. displaceable) or noncompetitive[33]. The IC50 value is expected to increase linearly with S concentration for competitive inhibition[33]. IC50 values indeed yielded a linear regression with increasing concentration of S (Fig. 4d), indicating that both B07 and B09 inhibited S binding through a competitive mechanism.

## Structural characterization of the VHHs /sACE2 interaction

We undertook structural studies to define the epitopes of the VHHs and the inhibitory mechanism of B07. We obtained crystals of a ternary complex of the soluble, monomeric protease domain of hACE2 and VHHs B07 and B10 that diffracted at 2.7 Å resolution (Fig. 5a, b, Supplementary Table 2, Supplementary Fig. 4a, Supplementary Fig. 5). The X-ray structure revealed that B10 buries, from the surrounding solvent, an area of 601 Å² of the hACE2 surface (Supplementary Fig. 4b, c), recognizing a negatively charged surface in loops β1/β2 and helices α5/6/11 (Fig. 5a, Supplementary Fig. 4d, e, Supplementary Fig. 6, Supplementary Table 3), away from the RBD footprint (Supplementary Fig. 7a–c), explaining why B10 did not inhibit SARS-CoV-2 infection. We validated the epitope by introducing the E160A mutation in hACE2 helix α6, thereby breaking the negatively charged patch. As predicted, E160A mutant did not bind B10 (Supplementary Fig. 4f). B07 binds mainly to hACE2 helix α1, burying a surface area of 662.1 Å² from solvent (Fig. 5c). It forms multiple polar and van der Waals interactions via the CDR1, 2, and 3 of B07 and hACE2 helix α1 (Fig. 5c−e, Supplementary Table 4, Supplementary Fig. 6). Residues from hACE2 helix α2 and loop β4/β5 (residue 353) also contribute to the interaction (Fig. 5c, e; Supplementary Table 4, Supplementary Fig. 6). From the 6 amino acid differences between B07 and B09, only two of them are in the paratope region, and one of these was a change from Ile (germline) to Thr (B07) or Ser (B09). Except for position 60, in which a somatic mutation from Ala to Pro took place only in B07, the remaining eight somatic mutations in the paratope gave rise to the same substitution in both VHHs (Supplementary Fig. 1), strongly indicating B07 and B09 recognize the same epitope. We validated the structure and B07 epitope by substituting the critical residues H34, E35, and D38 (Fig. 5f, Supplementary Fig. 8a). The respective mutants (H34A, E35A, D38A) abrogated B07 binding on hACE2 expressing cells (Fig. 5f), breaking a salt bridge (E35A) or hydrogen bonds (H34A, D38A).

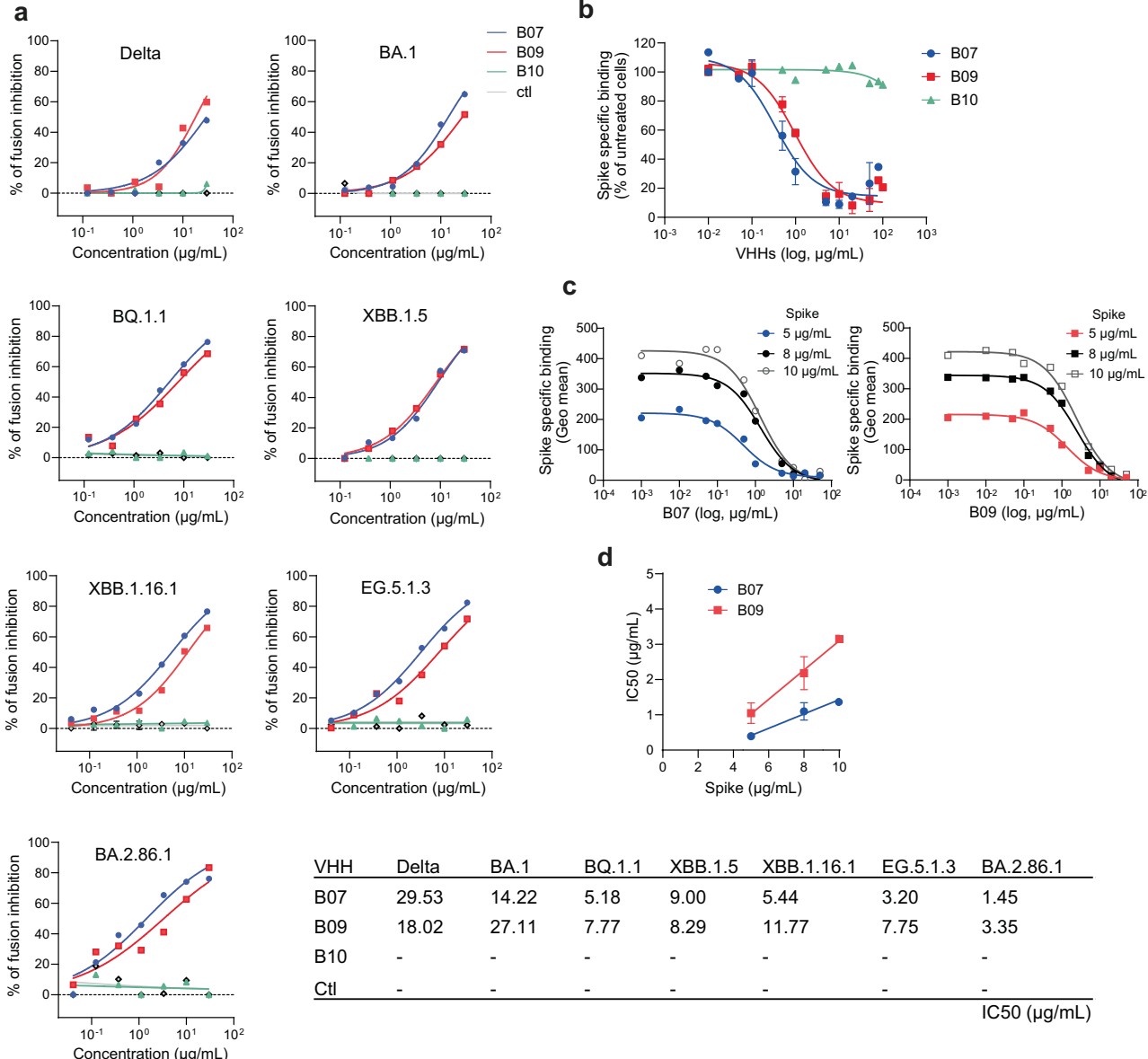

| VHH | Delta | BA.1 | BQ.1.1 | XBB.1.5 | XBB.1.16.1 | EG.5.1.3 | BA.2.86.1 |
|-----|-------|------|--------|---------|------------|----------|-----------|
| B07 | 29.53 | 14.22 | 5.18 | 9.00 | 5.44 | 3.20 | 1.45 |
| B09 | 18.02 | 27.11 | 7.77 | 8.29 | 11.77 | 7.75 | 3.35 |
| B10 | - | - | - | - | - | - | - |
| Ctl | - | - | - | - | - | - | - |

IC50 (µg/mL)

**Fig. 4 | Inhibition of SARS-CoV-2 induced fusion and of spike binding on hACE2 expressing cells by anti-ACE2 VHHs. a** S-fuse assay. Inhibition by B07, B09, B10 VHHs after infection of S-Fuse cells (U2OS-hACE2 GFP1-10 and GFP11) with different SARS-CoV-2 variants (Delta B.1.617.2, BA.1, BQ.1.1, XBB.1.5, XBB1.16.1, EG.5.1.3, BA.2.86.1). Data are mean of two independent experiments. The dashed line indicates the limit of detection. An anti-IgM (or IgE) specific VHH was used as a control (ctl). IC50 values are indicated in the table. **b** Inhibition of 5 µg/mL spike binding to hACE2-expressing HEK293 cells by increasing concentrations of VHHs B07, B09 or B10. HEK293-hACE2 cells pretreated or not with the VHHs and incubated with soluble spike (S) protein (ancestral spike) were stained with an anti-S antibody. Results were normalized for nonspecific (0%) and specific binding in the absence of inhibitor (100%). Experiments were fitted to a one-site competitive binding model. Data are mean ± SD of three (B07, B09) independent experiments or mean of two independent experiments (B10). **c** Displacement of S (ancestral spike) binding by B07 (left) or B09 (right) for various concentrations of S (5 µg/mL, 8 µg/mL, 10 µg/mL). Experiments were carried out as in (**b**) (one representative experiment out of two - 10 µg/mL, or three - 5 µg/mL, 8 µg/mL). **d** Relationship between the observed IC50 values for B07 or B09 displacement of S binding and initial concentration of S (ancestral spike). Results represent means ± SD of three independent experiments (5 µg/mL, 8 µg/mL) or means of two independent experiments (10 µg/mL). Linear regression analysis: R² B07 = 0.9005; R² B09 = 0.7896. Source Data are provided as a Source Data file.

These mutants also inhibited the binding of B09 with cell surface hACE2 (Supplementary Fig. 8b), corroborating that B09 shares the epitope with B07. The calculated surface electrostatic potential indicated a strong charge complementarity (Fig. 5g). Super-imposition of the hACE2/B07/B10 complex model with the previously reported RBD/sACE2 model (PDB 6M0J) on hACE2 revealed steric clashes between B07 (purple) and the RBD of SARS-CoV-2 (green), when bound to their respective epitope (Fig. 5h, i, Supplementary Fig. 7a–c). This demonstrates that they cannot bind simultaneously and explains the ability of B07 to inhibit the interaction of the spike with hACE2. Nine residues (underlined in Supplementary Fig. 7b, in red in Supplementary Table 5) of hACE2 were involved in direct contact with both B07 and the RBD. These findings explained the results of competition assays between VHHs and S (Fig. 4b–d). Comparison of B07 epitope with the RBD footprints on ACE2 of various coronavirus using ACE2 as a receptor (including, in addition to the sarbecoviruses, the merbecovirus NVHKU5R[34] and the alpha-coronavirus HCoV-NL63[35]) (Supplementary Fig. 7d, e; Supplementary Table 5), revealed a substantial overlap. These observations indicate that B07 broadly competes with the binding of multiple coronaviruses. Finally, aligning the ACE2 amino acid sequences from different species showed that there is a phylogenetic diversity at

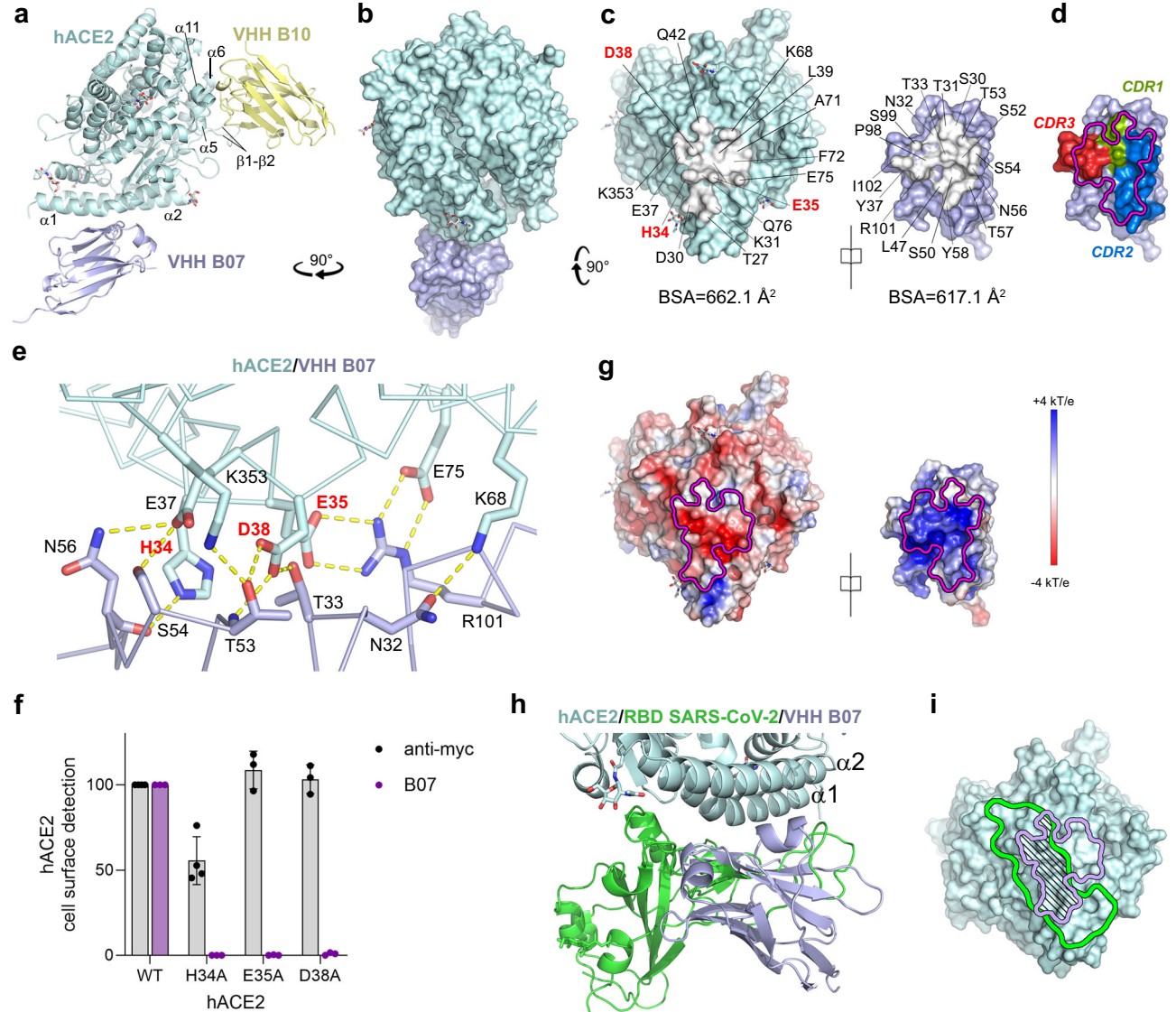

**Fig. 5 | X-ray structure of hACE2-B07-B10 complex explains the competitive binding of B07 and RBDs on hACE2. a** Crystal structure of hACE2 (light blue) in complex with B07 (purple) and B10 (yellow) (PDB: 9R19). **b** Surface representation of the complex between hACE2 (light blue) and B07 (purple) rotated to 90 °C from **a**. For clarity B10 was not displayed. **c** Open-book representation and footprints (in white) of B07 on hACE2 (left) and hACE2 on B07 (right). The buried surfaces (BSA) are indicated under each surface. Residues of hACE2 in red correspond to those for which alanine substitutions compromise B07 VHH binding to hACE2 mutants (**f**). **d** B07 complementarity determining regions (CDRs) colored in green (CDR1), blue (CDR2), and red (CDR3). hACE2 footprint on B07 is represented. **e** Hydrogen bonds and salt bridges (yellow dashed lines) between hACE2 and B07. **f** Impact of hACE2 alanine substitution on B07 binding. Cells were transfected with myc tagged or untagged hACE2 alanine mutants, incubated with an anti-myc antibody or B07 (10 μg/mL), respectively and stained with a mouse anti-myc antibody and a AF647-conjugated anti-mouse antibody before being analyzed by flow cytometry. Data are mean ± SD of four (anti-myc) or three (B07) independent experiments. Source Data are provided as a Source Data file. **g** Electrostatic potential mapped on the surface of the structure of hACE2 and B07. **h** Superimposition of the binding area of B07 (purple) and SARS-CoV-2 RBD (PDB: 6M0J, green) on hACE2 (light blue). **i** Footprints of the site of interaction of B07 and RBD on hACE2. The contours are colored in purple (B07) or in green (RBD). The common area is striped.

ACE2 positions 30, 31, 34, 35, 37 (Supplementary Fig. 6), suggesting that B07 may be impaired in its binding ability with hamster and mouse ACE2.

### VHH B07 dimerization potentiates its ability to inhibit hACE2-mediated coronavirus entry

We explored the effect of bivalent VHH variant by making a dimeric form[27] with VHH B07 in fusion with the human IgG1 Fc segment (i.e. hinge plus CH2 and CH3 domains)[36]. We tested the resulting B07-Fc for its ability to (i) interact with soluble hACE2 in vitro, (ii) bind membrane ACE2 using different cell types, (iii) prevent the enzymatic activity of hACE2, and (iv) inhibit SARS-CoV-2 variants (Fig. 6a–h).

B07-Fc bound soluble hACE2 with higher avidity (Fig. 6a) compared to monomeric B07 (Fig. 1), with a measured apparent $K_D$ ($K_{D\ App}$) of 0.11 nM. B07-Fc was specifically detected at the surface of different cell lines expressing exogenous (HEK293-hACE2, A549-hACE2) or endogenous ACE2 (IGROV-1, Vero E6) by flow cytometry after staining with anti-IgG antibodies (Fig. 6b, c, Supplementary Fig. 9a, b). B07-Fc also specifically stained primary human nasal epithelial cells (hNECs), as shown by confocal immunofluorescence imaging (Fig. 6d). This air-liquid culture system is an effective tool for studying SARS-CoV-2 infection[37]. As described[38,39], hACE2 co-localized with α-tubulin on the surface of motile cilia (Supplementary Fig. 9c). As seen with B07, B07-Fc did not affect enzymatic peptide cleavage activity of soluble and cell

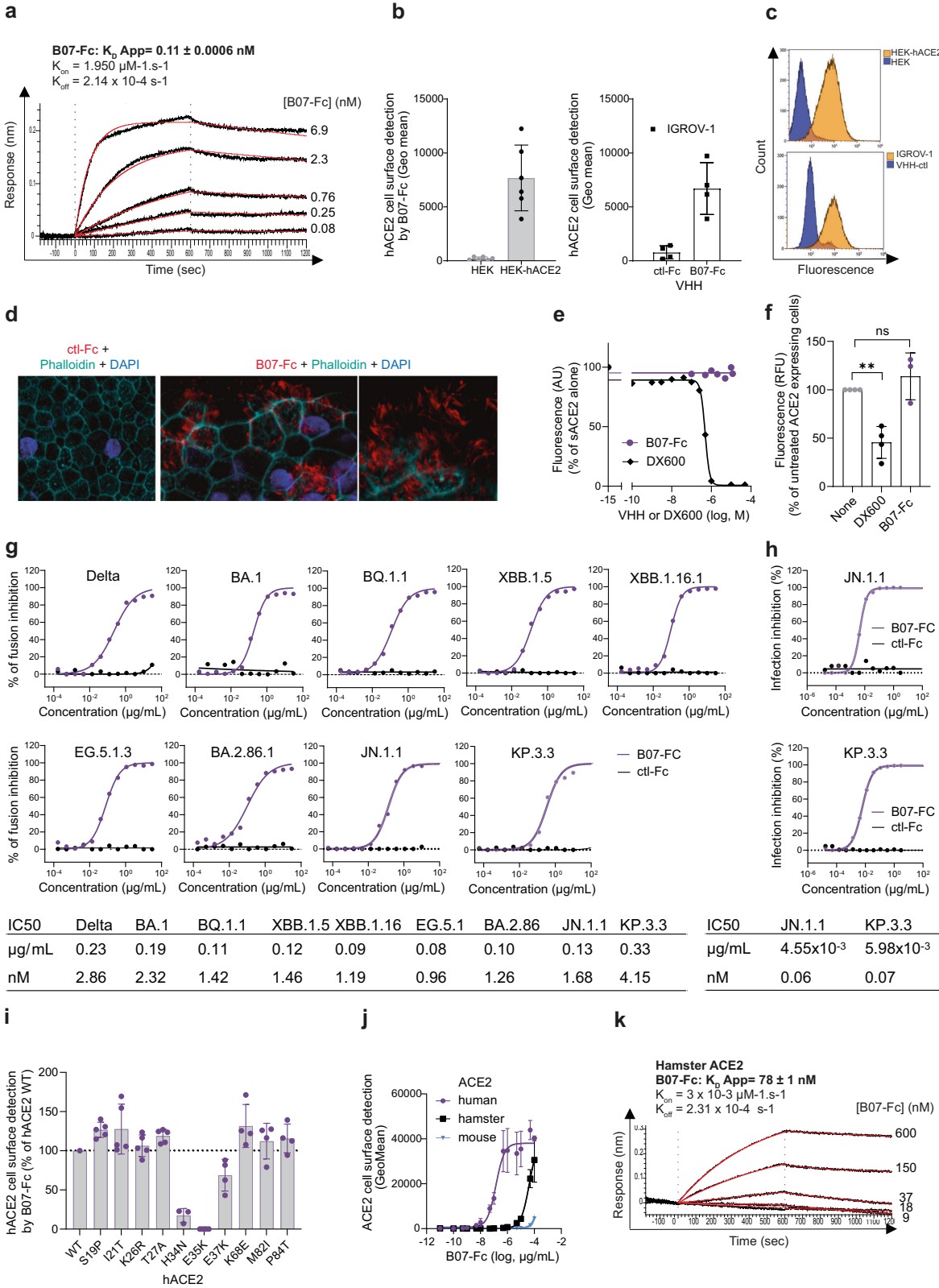

| IC50 | Delta | BA.1 | BQ.1.1 | XBB.1.5 | XBB.1.16 | EG.5.1 | BA.2.86 | JN.1.1 | KP.3.3 |
|---|---|---|---|---|---|---|---|---|---|
| µg/mL | 0.23 | 0.19 | 0.11 | 0.12 | 0.09 | 0.08 | 0.10 | 0.13 | 0.33 |
| nM | 2.86 | 2.32 | 1.42 | 1.46 | 1.19 | 0.96 | 1.26 | 1.68 | 4.15 |

| IC50 | JN.1.1 | KP.3.3 |
|---|---|---|
| µg/mL | 4.55x10⁻³ | 5.98x10⁻³ |
| nM | 0.06 | 0.07 |

surface hACE2 (Fig. 6e, f) and did not impact cell viability, even at high concentration (Supplementary Fig. 9d). Finally, B07-Fc inhibited syncytia formation induced with the 9 SARS-CoV-2 variants and subvariants tested (Delta, BA.1, BQ.1.1, XBB.1.5, XBB.1.16.1, EG.5.1.3, BA.2.86.1, JN.1.1, KP.3.3)[5] (Fig. 6g, Supplementary Table 1c) with an IC50 at least 2 log lower than that of B07 (IC50 $_{B07-Fc}$ = from 0.96 to 4.15 nM versus IC50 $_{B07}$ = from 95.1 to 1937.2 nM). B07-Fc also inhibited infection of Delta and Omicron variants as measured by viral N protein detection (IC50 $_{B07-Fc}$ = 0.06 and 0.07 nM for JN1.1 and KP.3.3, respectively) (Fig. 6h, Supplementary Table 1c) and by RT-qPCR (IC50 $_{B07-Fc}$ from 7.9−22.4 nM for Delta, BA.1, BQ.1.1, XBB.1.5, XBB.1.16.1) (Supplementary Fig. 9e). These results suggested that the dimerization of the VHH B07 increased its inhibitory efficacy without affecting hACE2 enzymatic activity. Therefore, this dimeric VHH represented a good candidate for further in vivo studies.

**Fig. 6 | Fusion of VHH B07 to the Fc fragment enhances its avidity and potency.**
**a** Kinetic analysis of VHH B07-Fc on hACE2-His by BioLayer Interferometry (BLI) using various concentrations of VHH B07-Fc (indicated on the figure). HIS2 biosensors were used to immobilize 1 μg/mL of hACE2-His. **b** VHH B07-Fc binding on cells expressing exogenous (HEK293-hACE2) or endogenous hACE2 (IGROV-1). Experiments were performed as in Fig. 2 but using 0.1 μg/mL of VHH B07-Fc. Data are mean ± SD of six (HEK293) or four (IGROV-1) independent experiments.
**c** Fluorescence diagram overlays as performed in Fig. 2c. **d** hACE2, phalloidin (F-actin) and DAPI staining on primary human nasal epithelial cells (hNECs). Representative immunofluorescence staining (out of 4 independent experiments) of hACE2 (Red: B07-Fc staining) in combination with phalloidin (Cyan) and DAPI (Blue). Scale bars: ctl-Fc, 10 μm; B07-Fc, 5 μm. **e** Enzymatic activity of soluble hACE2 (sACE2) in the presence of VHH B07-Fc or absence using the SensoLyte 390 ACE2 Activity Assay Kit, as performed in Fig. 3. DX600 was an inhibitor control of the kit (one representative experiment out of two; mean values of duplicates). **f** Cell surface hACE2 activity in the presence of saturating concentration of VHHs (10 μM) or 1 μM DX600. Data are means ± SD of three independent experiments. Paired T test two-sided: ns, nonsignificant; ** $p = 0.0070$. **g** Inhibition of fusion (S-Fuse assay) by B07-Fc after infection of S-Fuse cells with different SARS-CoV-2 variants: Delta

B.1.617.2, BA.1, BQ1.1, XBB.1.5, XBB.1.16.1, EG.5.1.3, BA.2.86.1, JN.1.1, KP.3.3. The dashed line indicates the limit of detection. Data are mean of two independent experiments, except for BQ1.1 (three experiments). **h** Inhibition of infection quantified by counting SARS-CoV-2 nucleoprotein (N) positive cells. IGROV-1 cells were pre-incubated 1 h with serial dilutions of B07-Fc and infected with the indicated variants at $3 \times 10^{-2}$ infectious units per cell. Cells were stained with a pan-coronavirus anti-N antibody at day 1 pi. The percentage of inhibition is represented. One representative experiment performed in duplicates. **i–k** Impact of ACE2 genetic variation on VHH B07-Fc binding. **i, j** Cells were transfected with different ACE2 constructs (**i**) hACE2 polymorphism; (**j**) human, hamster, or mouse ACE2, incubated with B07-Fc (0.1 μg/mL or different doses) and stained with an AF647-conjugated anti-human antibody before being analyzed by flow cytometry. **i** Data are mean ± SD of independent experiments: $n = 5$ (S19P; I21T; K26R; T27A; E35K) or $n = 4$ (E37K; K68E; M82I, P84T) or $n = 3$ (H34N). **j** Data are mean ± SD of 5 (human ACE2) or 3 (hamster and mouse ACE2) independent experiments. **k** Kinetic analysis of B07-Fc on hamster ACE2-His by BioLayer Interferometry (BLI) using various concentrations of VHH B07-Fc (indicated on the figure). HIS2 biosensors were used to immobilize 2 μg/mL of His-ACE2. $K_{D\,App}$ is indicated. Source Data are provided as a Source Data file.

## Genetic variation of the ACE2 N-terminal region impacts VHH B07-Fc binding efficacy

To evaluate whether human ACE (hACE2) polymorphisms affect B07-Fc binding efficacy, we constructed polymorphic hACE2 variants by substituting residues located within the vicinity of S and B07 binding (S19P, I21T, K26R, T27A, H34N, E35K, E37K, K68E, M82I, P84T) in a myc-hACE2 expressing plasmid. Then, we tested their cell surface expression after anti-myc staining and their reactivity with B07-Fc by flow cytometry. None of the substitution abrogated cell surface expression of the myc-hACE2 constructs (Supplementary Fig. 10a). Eight out of ten constructs were detected by B07-Fc, suggesting that hACE2 polymorphism might have little impact on B07-Fc antiviral efficacy (Fig. 6i). However, B07-Fc failed to detect hACE2-H34N and hACE2-E35K expressing cells (Fig. 6i), confirming the importance of the residues H34 and E35 for B07-Fc binding to hACE2 (Fig. 5).

Phylogenetic diversities in ACE2 might also influence B07-Fc binding (Supplementary Fig. 6). B07-Fc poorly recognized hamster (haACE2) and mouse ACE2 expressing cells compared to hACE2 expressing cells (Fig. 6j). BLI experiments confirmed that B07-Fc bound poorly to the soluble hamster ACE2 compared to human ACE2 ($K_{D\,App}$ haACE2 = 78 nM versus $K_{D\,App}$ hACE2 = 0.11 nM) (Fig. 6k). The presence of a glutamine at position 34 (Q34) in these different species (Supplementary Fig. 6, Supplementary Fig. 10b) likely explain the impaired binding efficacy. Indeed, B07-Fc did not react with hACE2-H34Q (Supplementary Fig. 10c). In contrast, B07-Fc showed stronger binding for haACE2-Q34H than for haACE2-WT (Supplementary Fig. 10c). These results confirmed that B07-Fc epitope was critically dependent upon residues at position 34 of ACE2. Since residues at position 30-31 are also part of the epitope (Fig. 5c, Supplementary Table 4), it is likely that the presence of two asparagines at position 30-31 in mouse ACE2 (Supplementary Fig. 10b) explains the lower binding efficacy of B07-Fc to mouse ACE2 compared to hamster ACE2 (Fig. 6j).

## B07-Fc protects against SARS-CoV-2 infection in vivo

We evaluated in vivo the prophylactic potential of VHH B07-Fc using the hACE2 transgenic mouse model (K18-hACE2) and XBB.1.5 Omicron variant infection. Mice treated intranasally with a single administration of VHH B07-Fc (7 mg/kg), or VHH-Fc control (7 mg/kg) were infected 24 h later intranasally with $1 \times 10^5$ PFU of XBB.1.5 variant. Animals were euthanized at day 3 post-infection and lung viral load was measured by RT-qPCR and a conventional plaque assay (Fig. 7a–c).

In 6 of the 10 B07-Fc-treated mice, lung viral load level was strongly reduced ( > 3 log) compared with the mean of the control group or below the detection limit, suggesting that the lung had been protected from viral infection (Fig. 7b, c). We assessed by

immunofluorescence SARS-CoV-2 infection and the cytopathic effect induced by XBB.1.5 in the presence (B07-Fc) or absence (ctl-Fc) of B07-Fc (Fig. 7d). Nasal, respiratory, and lung epithelia were stained at day 3 p.i. with phalloidin (to visualize F-actin), anti-alpha tubulin antibodies (to visualize cilia), anti-SARS-CoV-2 N antibodies (to visualize viral replication), and DAPI (to visualize the nucleus). Treated mice showed no viral replication and no tissue damage (disappearance of the ciliated structure) compared to control mice (ctl-Fc and uninfected K18-hACE2) (Fig. 7d, Supplementary Fig. 11), corroborating the protective effect of B07-Fc. The dose of B07-Fc in the lung homogenate measured by ELISA was variable between treated mice, revealing probable heterogeneity in the biodistribution of VHH in the lung (Fig. 7e). Notably, all mice in which the B07-Fc dose in the lungs were above 215 ng B07-Fc/g of lung had lung viral load below the limit of detection, while the mice with lower B07-Fc dose exhibited high lung viral load (Fig. 7e). Therefore, we concluded that B07-Fc, when used prophylactically in K18-hACE2 and when efficiently delivered into the lungs, strongly protects against Omicron pulmonary infection.

To further confirm the protective effect of B07-Fc in vivo, we investigated the efficacy of B07-Fc in preventing lethality in K18-hACE2 mice infected with the D614G variant. Mice were treated intranasally with a single dose of B07-Fc (5 mg/kg) or VHH-ctl (5 mg/kg) and infected 24 h later with $1 \times 10^4$ PFU of the D614G variant. Body weight loss and signs of disease were measured starting from day 3 post-infection (p.i.) (Fig. 7f). All mice in the ctl-Fc group reached humane endpoints (body weight loss > 20%, clinical score > 6) at day 6 p.i. and were euthanized (Fig. 7g). Mice in the B07-Fc-treated group were also sacrificed on day 6 p.i. for measuring B07-Fc concentration in the lung as in Fig. 7e (Fig. 7h). All treated mice showed delayed body weight loss and pathological scoring, although this delay was variable between treated mice (Fig. 7g). The four mice with B07-Fc concentrations in the lung homogenate reaching > 200 ng B07-Fc/g of lung (VHH_high group) showed moderate symptoms with weight loss < 10%. On the other hand, mice with a low B07-Fc concentration in the lung (VHH_low group) experienced body weight loss and symptoms similar to those of the ctl-Fc group (Fig. 7g, h). The body weight curve of the VHH_high group was significantly delayed compared with that of the ctl-Fc group ($p = 0.000396$ using a linear mixed model). These results suggest that B07-Fc was able to protect mice when efficiently delivered to the lung (above a threshold of around 200 ng B07-Fc/g of lung).

We also evaluated the prophylactic potential of VHH-B07-Fc using the Syrian hamster preclinical model. VHH-B07-Fc was administered intranasally at 5 mg/kg and infected 24 h later with $1 \times 10^4$ PFU of the XBB.1.16.1 variant. Lung viral load was modestly but significantly lower in hamster treated with B07-Fc compared with the control animals

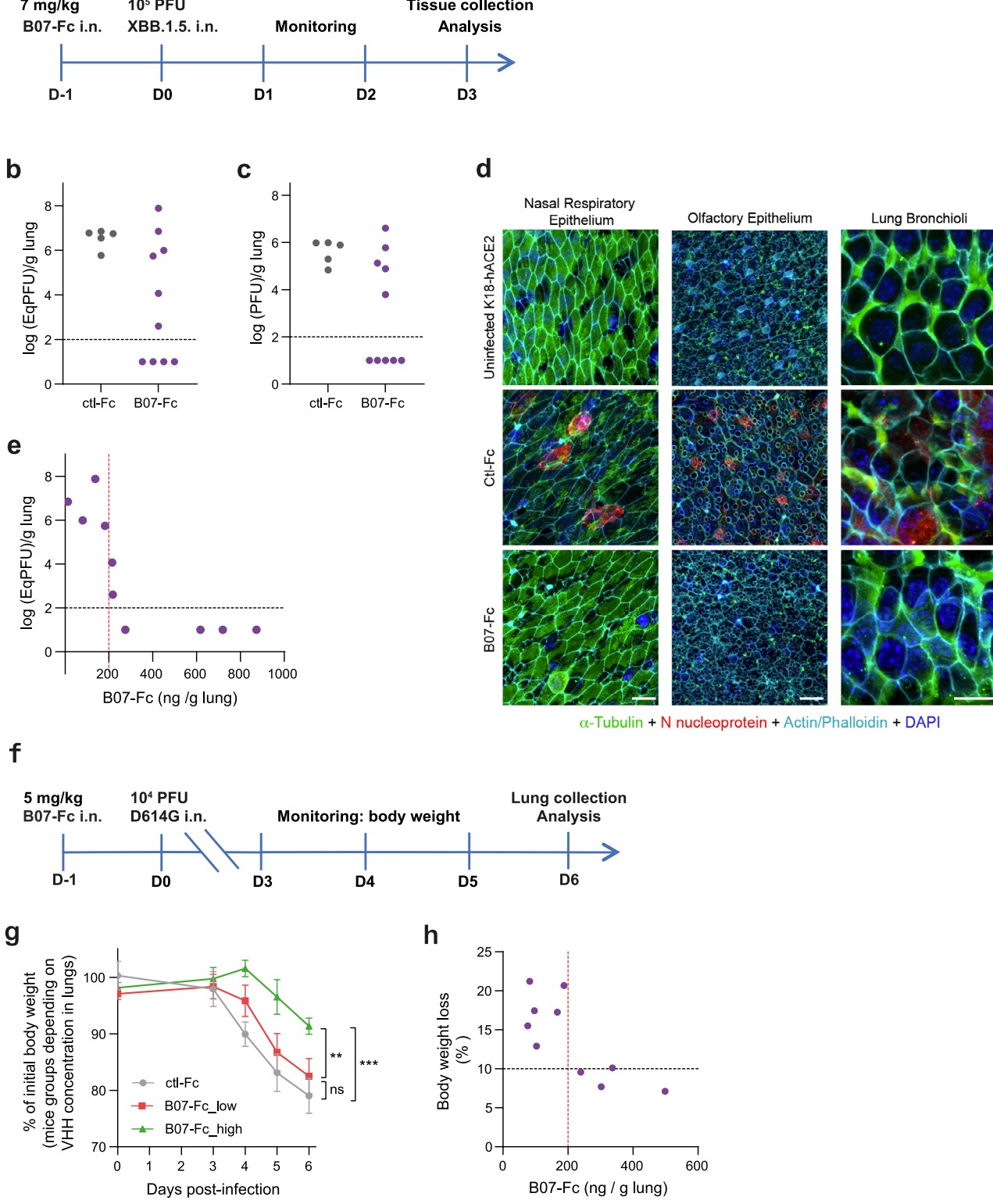

α-Tubulin + N nucleoprotein + Actin/Phalloidin + DAPI

(3.49 x 10⁷ vs 5.7 x 10⁸ copies/g lung) (Supplementary Fig. 12), indicating a lower protection potential of VHH-B07-Fc in this species compared with transgenic mice expressing human ACE2.

## Discussion

Anti-SARS-CoV-2 monoclonal antibodies (mAbs) used for prophylaxis or therapeutic treatment are losing their efficacy in the face of the continuous emergence of resistant variants[7]. Therefore, targeting ACE2 represents an alternative to the use of mAbs targeting the spike (S) protein. In line with previous studies[21–25], we found that anti-ACE2 conferred inhibition against a broad spectrum of variants (Figs. 4 and 6). We described here anti-hACE2 VHHs inhibiting infection by 9 variants, including Omicron JN.1.1 and KP.3.3 subvariants (Delta, BA.1, BQ1.1, XBB.1.5, XBB.1.16, EG.5.1, BA.2.86, JN.1.1, KP.3.3). In

**Fig. 7 | Prophylaxis of VHH B07-Fc for SARS-CoV-2 infection in mice. a** Schematic diagram showing the experimental design of B07-Fc prophylaxis in XBB.1.5 infected K18-hACE2 mice. Animals received intranasally (i.n.) 7 mg/kg VHH-B07-Fc (B07-Fc) or 7 mg/kg VHH-Fc ctl (ctl-Fc). Twenty-four hours later, they were infected with $10^5$ PFU XBB.1.5. SARS-CoV-2 intranasally (i.n.). Three days post-infection, different tissues were collected for analysis. **b** Genomic RNA load measured by RT-qPCR of SARS-CoV-2 in lung. **c** SARS-CoV-2 infectious particles measured by conventional plaque assays. **d** Tubulin, phalloidin (F-actin), SARS-CoV-2 nucleocapsid (N) (viral replication) and DAPI staining on nasal respiratory epithelium, olfactory epithelium, and lung bronchioli extracted from K18-hACE2 uninfected mice, and animals that received VHH-Fc (Ctl-Fc or B07-Fc). Representative immunofluorescence staining of Tubulin (Green) in combination with phalloidin (Cyan), SARS-CoV-2 N protein (Red), and DAPI (Blue). This was performed on nasal, olfactory, and pulmonary epithelium of two VHH ctl-treated mice and four B07-Fc treated mice. Scale bars: 10 μm. **e** Relationship between viral genomic RNA load measured in the lung by RT-qPCR and B07-Fc concentration quantified in the lung by ELISA. **f** Schematic diagram showing the experimental design of B07-Fc prophylaxis in D614G infected K18-hACE2 mice. Animals received intranasally (i.n.) 5 mg/kg VHH-B07-Fc (B07-Fc) or 5 mg/kg VHH-Fc ctl (ctl-Fc). Twenty-four hours later, they were infected with $10^4$ PFU D614G SARS-CoV-2 i.n. Mice were clinically monitored between days 3 to 6. Mice were euthanized on day 6 and lungs were collected for analysis. **g** Body weight measured over time post-infection in ctl-Fc ($n = 5$) and B07-Fc-treated groups. Mice of the B07-Fc group were split into VHH_high ($n = 4$) and VHH_low ($n = 6$) according to the B07-Fc concentration measured in the lung on day 6 p.i. (higher or lower than 200 ng/g of lung, respectively). Data are presented as mean values +/- SD. Linear mixed model two sided (no adjustment for multiple comparisons): ns, non-significant (p = 0.26); ** p = 0.00378; *** $p$ = 0.000396. **h** Relationship between body weight loss and B07-Fc concentration quantified in the lung by ELISA on day 6 p.i. Source Data are provided as a Source Data file.

comparison, the therapeutic anti-S mAb sotrovimab lost antiviral activity against BA.2.86.1 and JN.1[5,40]. Similarly, the mAbs pemivibart and sipavibart remained poorly active or lost efficacy against KP3.3[7], underscoring the pertinence of our host-targeting approach.

An issue with targeting host proteins is the potential loss of physiological functions of the target. Here, the anti-hACE2 VHHs bind the peptidase domain of hACE2 (Fig. 5) without interfering with its peptidase activity, as showed both in vitro and *in cellula* (Fig. 3, Fig. 6e, f). They inhibit SARS-CoV-2 infection by competitively blocking the interaction with S (Fig. 4b–d), targeting an epitope comprising residues mainly located in hACE2 helix α1 (Fig. 5), which are also recognized by the viral RDB (Supplementary Fig. 7). This epitope, distal to the enzyme's active site (Supplementary Fig. 6), should not induce side effects if it is targeted, supporting the translational potential of these antibodies. An additional advantage is that the emergence of escape mutants after treatment with competitive anti-ACE2 VHHs is unlikely.

Only low-frequency polymorphisms in hACE2 (H34N and E35K, $p \sim 10^{-5}$)[41] abolished B07-Fc binding, suggesting that variations in hACE2 should have little effect on the VHH efficacy. We note that these two hACE2 variants may also block RBD binding (Supplementary Fig. 7)[42]. Structural comparisons of the footprint of the RBDs of SARS-CoV and HCoV-NL63 revealed that B07-Fc is also expected to block these two related viruses, thus expanding the scope of its activity. We also compared the footprint of B07 on hACE2 with the one of NVHKU5R on mink ACE2. Indeed, mink has been identified as a reservoir for coronaviruses[43]. The epitope of NVHKU5R on mink ACE2 overlaps with the one of B07 on hACE2 (Supplementary Fig. 7d, e), suggesting that B07-Fc will also block potential emerging human-adapted coronavirus derived from mink.

Our VHHs have the capacity to rival previously described anti-hACE2[21–23]. B10, which is not inhibitory but binds to hACE2 with high affinity ($K_{D\ App} = 21$ nM), represents a valuable tool for tracking hACE2 in SARS-CoV-2 infected cells and/or detecting hACE2 in cryptic primary cell locations that are not accessible to conventional antibodies. B07 and B09 displayed moderate affinities for hACE2 ($K_D \sim 200$ nM) and limited potencies in inhibiting SARS-CoV-2 (IC50 ~ 100 to 1000 nM). The variation in IC50 values observed across variants (Fig. 4a) was likely attributed to inherent experimental bias; however, differences in how the various spikes engage hACE2 (i.e., binding affinity and/or positioning) may also influence their potency. B07-Fc exhibited high affinity for hACE2 ($K_{D\ App} = 0.11$ nM) and potent efficacy in inhibiting SARS-CoV-2 infection (IC50 ~ 1 nM corresponding to ~ 5 ng/mL). It is one order of magnitude more potent than the most promising mAbs anti-hACE2 05B04 (IC50 ~ 50 ng/mL), when evaluated in the same infection inhibition assay[22]. Both B07-Fc and 05B04 act by a competitive mechanism, engaging hACE2 residues also targeted by the RBD (residues 27-42 for B07-Fc; residues 19-31 for 05B04[22]) (Supplementary Fig. 7). Compared to conventional mAb-based therapies[21–23], VHH-

based therapies have several advantages. VHHs are smaller in size than mAbs, even when fused to a Fc fragment (~80 kDa for VHH-Fc vs ~150 kDa for IgG), and they show improved tissue penetration[26,27]. Their single-gene format is also simpler to engineer, reducing production cost. VHHs could be engineered into a multi-specific format by combining two or three VHHs in a single polypeptide to target distinct epitopes or even distinct receptors[44], potentially with synergistic effects[45]. We found that engrafting B07 onto the Fc domain of conventional IgG1 results per se in a dimeric VHH with highly increased inhibitory efficacy (Fig. 6g, h), while showing no toxicity on cells (Supplementary Fig. 9d). The fusion with conventional Fc domain allows VHHs to bind Fc receptors, thereby rescuing them from degradation in vivo[29], which improves their pharmacokinetic profile[27,28]. However, Fc fusion may activate Fc effector function in immune cells or with complement, or cause antibody-dependence enhancement (ADE)[46]. To mitigate these risks and increase the potential of therapeutic properties, an optimal balance of Fc effector functions can be achieved by introducing point mutations in the Fc domain[47,48]. It would be of interest to explore the effect of known mutations on the efficacy of VHH B07-Fc.

Without additional development in the formulation, VHH B07-Fc is already highly stable to be administered by inhalation[49,50]. B07-Fc prevented the infection of human ACE2-expressing mice (K18-hACE2) by D614G or Omicron XBB.1.5 following intranasal administration (Fig. 7). The observed heterogeneity in the number of fully protected mice correlated with the effective concentration of B07-Fc in the lung, likely reflecting a sub-optimal distribution of B07-Fc. Although the administration procedures require optimization to further boost overall protection[51], beyond a threshold dose of B07-Fc in the lung (about 200 ng of B07-Fc/g lung), no viral load was detected in animals. We confirmed this protective effect of B07-Fc after intranasal administration in the preclinical Syrian hamster model[52–54] (Supplementary Fig. 12), although this protection was reduced. The weak binding of B07-Fc to hamster ACE2 compared to hACE2 (Fig. 6k) likely explains the low efficacy of B07-Fc treatment in hamsters. Intranasal administration of anti-S VHHs has already proved its efficacy to prevent SARS-CoV-2 infection in animal models[55–57] and to reduce the viral load in the brain[58]. In the case of anti-hACE2, this mode of administration, by promoting bio-distribution in the lung, may also overcome a loss of anti-hACE2 activity due to the wide expression of hACE2 in different tissues and the presence of circulating sACE2 in the blood[15]. This feature could result in the use of much reduced dose to achieve antiviral activity. In addition, respiratory drug delivery presents low logistical burden and possible self-administration, which would be particularly beneficial in the case of SARS-CoV-2[55].

Overall, our anti-hACE2 VHHs provide a potential solution to the never-ending escape of SARS-CoV-2 from anti-S antibodies, offering novel opportunities for therapeutic treatments and prophylaxis.

## Methods

### Expression and purification of hACE2 peptidase domain

For alpaca immunization, the human ACE2 peptidase domain (residues 19-615, Uniprot: Q9BYF1) was cloned in pcDNA3.1. The signal peptide from the IgK chain (METDTLLLWVLLLWVPGSTG) was introduced at the N-terminus to ensure the protein secretion to the extracellular medium, and His(x8) and Strep tags were added in-tandem at the C-terminus, following a Thrombin cleavage site. For structural studies, the Strep tag was exchanged for the ybbR tag (DSLEFIASKLA).

The plasmid coding for hACE2-His-Strep was transiently transfected into Expi293F™ cells (Thermo Fischer) using FectoPro® DNA transfection reagent (PolyPlus) according to the manufacturer's instructions, and 5 μM kifunensine was added to the culture medium immediately after transfection. The cells were incubated 5 days at 37 °C and harvested by centrifugation. The supernatant was concentrated and recombinant hACE2 was purified by affinity chromatography on a Streptactin column (IBA) and by size-exclusion chromatography (SEC) on a Superdex200 column (Cytiva) equilibrated with 10 mM Tris, 100 mM NaCl, pH8. hACE2 was digested over-night at room temperature with Thrombin (1 U per 100 μg of recombinant protein) to remove the His and Strep tags. The proteolysis was stopped with 1 mM PMSF (phenylmethylsulfonyl fluoride) and the untagged hACE2 was recovered in the flow-through after injection in a Streptactin column. A final SEC was performed, after which untagged hACE2 was concentrated and stored at -80 °C. For VHH epitope mapping experiments using BLI, hACE2 peptidase domain fused to a human Fc from IgG[36] was used (sACE2-Fc). For the measurement of the affinity between VHH B07-Fc and sACE2 by interferometry (BLI), we used a His-tagged sACE2 (sACE2-His) from human or hamster species (R&D systems).

For structural studies, the hACE2-His-ybbR plasmid was transfected into Expi293F™ GnTI- cells (Thermo Fischer A14527) using FectoPro® DNA transfection reagent (PolyPlus) according to the manufacturer's instructions. The cells were incubated 5 days at 37 °C and harvested by centrifugation. The supernatant was concentrated and recombinant hACE2 was purified by IMAC on a His-Excel column. The eluted protein was concentrated, the buffer was exchanged (10 mM Tris-HCl, 100 mM NaCl, pH 8.0) and hACE2 was digested over-night at room temperature with Thrombin (1 U per 100 μg of recombinant protein). The cleavage was stopped with 1 mM PMSF (phenylmethylsulfonyl fluoride) and untagged hACE2 was recovered in the flow-through fraction after injection on the His-Excel column. The protein was concentrated and incubated over-night at room temperature with 1 U of endoglycosidase H in acidic buffer (NaAc 100 mM, NaCl 100 mM, pH 5.5). A final SEC was performed using a column equilibrated with 10 mM Tris-HCl, 100 mM NaCl, pH 8.0.

### Plasmids, site-directed mutagenesis, and transfection

Mouse ACE2 plasmids was purchased from addgene (pscALPSpuro-MmACE2, #158087). Hamster ACE2 (haACE2) pcDNA3.1 hygro plasmid was synthetized by GeneArt (Thermo Fisher). The construct human ACE2 (hACE2) was obtained by substitution of the sequence encoding mCardinal from mCardinal-C1 plasmid (addgene, #54799) with hACE2 from the pLenti6-attB-hACE2-BSD[32]. The construct myc-hACE2 was obtained by substitution of the sequence encoding mCardinal from mCardinal-C1 plasmid (addgene, #54799) with myc-hACE2 from the pCEP4-myc-ACE2 (addgene, #141185)[59]. Polymorphism substitutions and alanine substitutions were generated by site-directed mutagenesis using the Q5-site directed mutagenesis kit (New England Biolabs) according to the manufacturer's instructions. The hACE2 and haACE2 mutants were all verified by sequencing (Eurofins) using the primers: CMV-Fw 5′CGCAAATGGGCGGTAGGCGTG and either hACE2-300-Rv 5′CTGTGCATCCCAGGCCTG or haACE2-275-Rv 5′CTGTGCATCC-CAGGCCTG. For transient expression, HEK293 (2 x 10⁵) cells were cotransfected (lipofectamine 2000) with WT or mutant ACE2 vectors and with eGFP-N1 (Clontech, Palo Alto, Calif.), in a 5:1 ratio. The eGFP positive cells were analyzed by flow cytometry after immunostaining.

### Viruses

SARS-CoV-2 Delta (B.1.617.2; GISAID ID: EPI_ISL_2029113), D614G (hCoV-19/France/GE1973/2020; GISAID ID: EPI_ISL_414631), Omicron BA.1 (GISAID ID: EPI_ISL_6794907), BQ.1.1 (GISAID ID: EPI_ISL_15731523), and JN.1.1 (hCoV-19/France/HDF-IPP21391/2023) were previously described[5,40,60,61]. XBB.1.5 was isolated from a naso-pharyngeal swab of an individual of Hôpital Européen Georges Pompidou (HEGP; Assistance Publique, Hôpitaux de Paris) (GISAID ID: EPI_ISL_16353849)[62]. The laboratory of Virology of HEGP sequenced the swabs. The patient provided written informed consent for the use of the biological materials. The XBB.1.16.1 strain (hCoV-19/France/GES-IPP07712/2023), the EG.5.1.3 strain (hCoV-19/France/BRE-IPP15906/2023), and the BA.2.86.1 strain (hCoV-19/France/IDF-IPP17625/2023) were supplied by the National Reference Centre for Respiratory Viruses hosted by Institut Pasteur (Paris, France) and headed by Dr Etienne Simon-Lorière / Marianne Rameix-Welti[5]. The human sample from which strain hCoV-19/France/GES-IPP07712/2023 was isolated has been provided by Dr Vanessa COC-QUERELLE from Laboratory Deux Rives. The human sample from which strain hCoV-19/France/BRE-IPP15906/2023 was isolated has been provided by Dr F. Kerdavid from Laboratoire Alliance Anabio, Melesse. The human sample from which hCoV-19/France/IDF-IPP17625/2023 was isolated was isolated by Dr Aude LESENNE from Cerballiance, Lisses (France). All strains were previously collected. Titration of viral stocks was performed on Vero E6 or IGROV-1 cells, with a limiting dilution technique enabling the calculation of the median tissue culture infectious dose or on S-Fuse cells. KP.3.3 strains (hCoV-19/France/BFC-IPP06087/2024) were isolated and amplified by the Reference center for Respiratory Viruses hosted by Institut Pasteur[7].

In replication experiments, XBB.1.16.1 (hCoV-19/France/GES-IPP07712/2023), XBB.1.5 (hCoV-19/France/PDL-IPP58867/2022), BQ.1.1 (hCoV-19/France/IDF-IPP50823/2022) and BA.1 (hCoV-19/France/PDL-IPP46934i/2021; GISAID ID: EPI_ISL_8353353) were supplied by the National Reference Centre for Respiratory Viruses hosted by Institut Pasteur (Paris, France). The human samples from which strains were isolated has been provided from the Laboratoire Deux Rives, CH Laval and Laboratoire des centres de santé et d'hôpitaux d'IDF. All strains were previously collected. Viral productions were performed on Vero-TMPRSS2 cells and titration of viral stocks was performed on Vero E6 cells. Viral concentrations were determined through Plaque Assay and the titers were quantified as plaque forming units (pfu) per milliliter.

The sequence of the viral stocks was verified by RNAseq by the National Reference Centre for Respiratory Viruses hosted by Institut Pasteur (Paris, France). All work with infectious virus was performed in biosafety level 3 containment laboratories at Institut Pasteur.

### Cell lines

HEK293-hACE2 (human embryonic kidney cell line, ATCC CLR 1573) were generated by transfection and cultured in Blasticidin (10 μg/mL). U2OS (human osteosarcoma cell line, ATCC Cat# HTB-96) and A549 (human lung epithelial cell line, ATCC cat# CCL-185) cells stably expressing hACE2, and U2OS-hACE2 cells stably expressing the GFP split system (GFP1-10 and GFP11; S-Fuse cells) were previously described[32]. These cells were cultured in DMEM or in F-12K Nutrient Mixture Media (A549) supplemented with 10% fetal bovine serum (FBS), and 1% penicillin/streptomycin. Blasticidin (10 μg/mL) and puromycin (1 μg/mL) were used to select for hACE2 and GFP split transgenes expression, respectively.

Vero E6 (Vero 76, clone E6, Vero E6, ATCC® CRL-1586TM, simian kidney cell line) was obtained from ATCC (USA) and cultured in DMEM medium (Gibco) containing 10 % (v/v) FBS (Gibco) and penicillin/streptomycin (Thermo Fisher Scientific). IGROV-1 cells (Human ovarian

cancer cell line) were from the NCI-60 cell line panel[40] and cultured in RPMI medium (Thermo Fisher) containing 10% FBS and penicillin/streptomycin. Vero E6 and IGROV-1 endogenously express ACE2.

Absence of mycoplasma contamination was confirmed in all cell lines with the Mycoalert Mycoplasma Detection Kit (Lonza). All cell lines were cultured at 37° C and 5 % $CO_2$.

## Animal immunization and library construction

All immunization processes were executed according to the French legislation and in compliance with the European Communities Council Directives (2010/63/UE, French Law 2013-118, February 6, 2013). The Animal Experimentation Ethics Committee of Pasteur Institute (CETEA 89) approved this study (2020-27412). We subcutaneously injected an adult alpaca at days 0, 21, and 36 with approximately 150 µg of sACE2 mixed with Freund complete adjuvant for the first immunization and with Freund incomplete adjuvant for the following immunizations. A blood sample of about 150 mL of the immunized animal was collected and Peripheral Blood Mononuclear cells (PBMCs) were isolated on a ficoll discontinuous gradient.

Briefly, the *vhh*-encoding genes isolated from PBMCs (approximately $2 \times 10^8$) were then cloned into the phagemid vector pHEN6 by using primers contained enzymatic SfiI and NotI restriction sites at the 5' and 3' ends, respectively. The size of the library was estimated at about $8 \times 10^7$ cfu (colony-forming units). Phage display protocol was performed as described in Lafaye et al.[63]. About $10^{12}$ phage-VHH diluted in PBS were used to perform three rounds of panning by using sACE2 coated on Nunc Immunotubes (Maxisorp) tubes (10 µg/mL). To increase the stringency, different blocking buffers for each panning were used: 2% skimmed milk as saturating agent for the first round; 5% BSA (Bovine Serum Albumin) and Odyssey blocking buffer (LI-COR Biosciences) diluted at 1/2 for the second and third rounds.

Following panning, individual clones were screened by standard phage-ELISA procedures using an HRP/anti-M13 monoclonal antibody conjugate (11973-MM05T). Clones were selected and their VHH-encoding DNA was sequenced by Eurofins Genomics (Ebersberg, Germany). Sequence analysis was conducted using SnapGene v8.1.0 and multiple sequence alignment with ClustalW2 (EMBL-EBI). Out of the 192 individual clones tested in the phage ELISA, 123 were positive. All 123 positive clones were sequenced, revealing that 80 belonged to a dominant cluster derived from the same *IGHV3SS3* gene as B07 and B09. Within this cluster: B07 was found 31 times; B09 was found 23 times. Given that B07 and B09 originate from the same *IGHV3SS3* gene and differ by only six amino acid mutations in their sequences and 2 in their CDRs, they likely represent variants of the same lineage. B10, in contrast, was found only once and originates from a different *IGHV3-3* gene, indicating a distinct evolutionary origin.

## VHH production

The pHEN6 vector allows the periplasmic expression and purification of selected VHHs with a c-myc tag and His-tag at the C terminal end in frame with the VHH. Transformed *E. coli* TG1 cells expressed VHHs in the periplasm after induction with IPTG (1 mM) for 16 h at 30° C. After centrifugation, cells were resuspended in PBS then lysed by polymixin B sulfate (1 mg/mL) for 1 h at 4° C, the periplasmic extract being obtained after centrifugation. Purified VHHs were obtained by His pure Cobalt sepharose beads (Thermo) according to the manufacturer's instructions.

## Dimeric VHH-Fc production

The coding sequences of the selected VHHs were digested with NcoI and NotI and subcloned into pFuse-huIgG-Fc2 digested with NcoI and NotI[64]. *E. coli* XL1-Blue transformants were obtained on 2YT agar plates containing zeocin 25 µg/mL. The plasmids coding for the recombinant proteins were purified with Nucleobond Xtra Midi Plus EF (Macherey Nagel), transiently transfected in Expi293™ cells (Thermo Fisher Scientific) using Fectro PRO DNA transfection reagent (Polyplus), according to the manufacturer's instructions. Cells were incubated at 37° C for 5 days and then the cultures were centrifuged. Proteins were purified from the supernatants by affinity chromatography using a HiTrap protein A HP (Cytiva), followed by Size Exclusion Chromatography on a Superdex 75 column (Cytiva) equilibrated in PBS. Peaks corresponding to the dimeric VHH-Fc proteins were concentrated and stored at -80° C until used.

## Biolayer interferometry (BLI)

Equilibrium binding was performed via BLI-analysis using an Octet HTX instrument (Sartorius). hACE2-Fc (or VHH-His) diluted into Sartorius's PBS working buffer containing a final concentration of 300 mM NaCl (pH 7.2) were immobilized onto Anti-hIgG Fc Capture (AHC) (or Ni-NTA) biosensors (Sartorius), which were pre-equilibrated in the same buffer. Following a baseline, ACE2-coated sensors (or VHH-coated sensors) were incubated with various concentrations of monomeric VHHs (or various concentration of soluble hACE2) for 10 min to reach equilibrium. Subsequently, biosensors were put in working buffer for 10 min to initiate dissociation. All incubations were performed at a temperature of 30° C under continuous shaking (1,000 rpm). To minimize classical surface-dependent avidity artefacts and avoid artificial multiphasic behavior, all binding experiments were performed using a range of protein loading concentrations. Optimal ACE2-Fc densities on the biosensor were determined to be 1 µg/mL for B07, B09, and B10. After minimizing avidity effects, binding data were analyzed using the Octet Software 13.0 and association/dissociation curves were globally fitted with a 1:1 binding model to determine $K_D$ apparent values[65]. For B07-Fc binding of human or hamster ACE2, sACE2-His (1 or 2 µg/mL, R&D Systems)-coated HIS2 sensors (Sartorius) were incubated with various concentrations of VHH-B07-Fc (as indicated) in 50 mM Tris-HCl buffer (pH 7.5), supplemented with 0.01% P20 and 0.1% BSA. Binding curves from the association and dissociation steps were fitted using a 1:1 binding model.

To measure competitive binding to sACE2 protein between the different VHHs by BLI, we immobilized hACE2 fused to the human Fc domain (sACE2-Fc) onto Anti-hIgG Fc Capture (AHC) Biosensors (Sartorius) at 5 µg/mL for 10 minutes. A baseline step was realized by dipping sensors into the working buffer. Then, a first VHH was applied at 5 µg/mL, in order to bind the immobilized hACE2 and reach saturation. The sensor was then dipped into a well containing a mixture of the same VHH with the competitor one added at the same concentration (5 µg/mL).

## Measurement of VHH binding or spike binding on ACE2 expressing cells by flow cytometry

Cells ($2 \times 10^5$) (transfected or not) were incubated for 60 min at 4° C with 10 µg/mL VHHs or 0.1 µg/mL VHH-Fc, washed twice with 2% FBS-PBS, incubated for 60 min at 4° C with anti-myc antibody (9E10, Abnova MAB0957), washed twice with 2% FBS-PBS, and incubated for 60 min at 4° C with AF488 (or AF647)-conjugated anti-mouse antibody (Invitrogen, A11029, A21236) (for VHHs detection), or with the AF488 (or AF647)-conjugated anti-human IgG antibody (Thermo Fischer A11013, A21445) (for VHH-Fc detection). The stained cells were washed twice, fixed in PFA 1.5% and analyzed on Attune flow cytometer NxT flow cytometer (Thermo Fisher) and Kaluza software v3.2.1. The parental cells (HEK293, A549) or a VHH control (anti-IgE or anti-IgM or R3VQ-Fc VHHs) were used as negative controls.

To perform competition experiments of soluble spike binding an increasing concentration of VHH (between 0.01 to 100 µg/mL) were incubated with HEK293-hACE2 cells in RPMI/10% FBS. After 30 min at 37 ° C, 5 µg/mL, or different doses (5, 8, 10 µg/mL), of soluble trimeric spike (ancestral spike)[66] was added for 30 min at 37° C. The "ancestral spike" referred here to SARS-CoV-2 Spike corresponding to the

primary isolate hCoV-19/France/IDF-0372/2020 (GISAID ID: EPI_ISL_406596), Pango lineage B, Nextstrain Clade 19 A. Then, immuno-staining with 5 µg/mL anti-spike monoclonal antibody mAb48 (gift of Hugo Mouquet, Institut Pasteur)[67] and alexa-Fluor-488-conjugated anti-human IgG antibody (Thermo Fischer A11013) was performed at 4° C. Fixed cells (PFA 1.5%) were analyzed on Attune flow cytometer NxT flow cytometer (Thermo Fisher) and analyzed as described above.

## ACE2 enzymatic activity

Human ACE2 activity was evaluated using SensoLyte 390 ACE2 Activity Assay Kit (AnaSpec) according to the manufacturer's protocol. To measure hACE2 activity on purified sACE2, 10 nM of sACE2 was incubated with B07, B09, and B10 at different concentrations before adding the fluorogenic peptide substrate, MethoxyCoumarin acetic acid-Ala (Mc-Ala)/Dnp, where Mc-Ala is quenched by Dnp. The protease activity of sACE2 is indicated by an increase of fluorescence after peptide cleavage, and Mc-Ala recovery, monitored at Ex/Em = 330 nm/390 nm (Tecan SPARK).

To measure hACE2 activity on cells, $2.5 \times 10^4$ HEK293-hACE2 cells per well (96-well black plate, Greiner-BioOne) were incubated with B07, B09, B10 (>20 µM) or the control inhibitor DX600 (1 µM) in 40 µL complete medium without phenol red (Gibco) before adding 40 µL of substrat (Mc-Ala/Dnp) diluted in PBS (SensoLyte 390 ACE2 Activity Assay kit). Peptide cleavage was measured in the supernatant on Tecan SPARK plate reader (SparkControl Magellan 3.2).

## SARS-CoV-2 inhibition assay: S-Fuse

U2OS-ACE2 GFP1-10 and GFP11 cells, also termed S-Fuse cells, become GFP+ when they are productively infected by SARS-CoV-2[32]. Cells were mixed (ratio 1:1) (U2OS-ACE2 GFP1-10 ($10^4$) and GFP11 cells ($10^4$)) and plated in a µClear 96-well plate (Greiner Bio-One). Serial dilution of VHHs were incubated with the cells at 37° C and infected with a SARS-CoV-2 strain. Eighteen hours later, cells were fixed with 4% paraformaldehyde (PFA), washed and stained with Hoechst (dilution 1:10,000, Invitrogen). The extent of fusion was then quantified by measuring the number of objects GFP+ with an Opera Phenix high-content confocal microscope (PerkinElmer) and Harmony software version 4.9 (PerkinElmer). The percentage of inhibition was calculated using the number of syncytia as value with the following formula: $100 \times (1 - (\text{value with VHH} - \text{value in 'non-infected'})/(\text{value in 'no VHH'} - \text{value in 'non-infected'}))$. Inhibition activity of each VHH was expressed as the IC50. We previously reported that the neutralization assay with the S-Fuse system is not affected by differences in fusogenicity between variants[68].

## SARS-CoV-2 inhibition assay: Quantification of SARS-CoV-2 nucleoprotein positive cells

Sixteen hours before infection, $3 \times 10^4$ IGROV-1 cells per well were seeded in a µClear black 96-well plate (Greiner Bio-One). Cells were incubated with serially diluted VHH-Fc for 1 h at 37° C and infected with the indicated SARS-CoV-2 strains. After 24 hours, cells were fixed with 4% PFA (Electron Microscopy Sciences, cat# 15714-S). The cells were then intracellularly stained with anti-SARS-CoV-2 nucleoprotein (N) antibody NCP-1 (0.1 µg/mL) as described[7]. The staining was carried out in PBS with 0.05% saponin 1% BSA, and 0.05% sodium azide for 1 hour. Cells were then washed twice with PBS and stained with anti-IgG Alexa Fluor 488 (dilution 1:500, Invitrogen; cat# A11029) for 30 minutes before being washed twice with PBS. Hoechst 33342 (Invitrogen, cat# H3570) was added during the final PBS wash. Images were captured using an Opera Phenix high-content confocal microscope (PerkinElmer). The N-positive area and the number of nuclei were quantified using Harmony Software v4.9 (PerkinElmer). The percentage of inhibition of infection was calculated using the N-positive area with the following formula: $100 \times (1 - [\text{value with VHH} - \text{value in "non-}$

infected"]/[value in "no VHH" − value in "non-infected"]). The half-maximal inhibitory concentration (IC50) was calculated with a reconstructed curve using the percentage of inhibition at each concentration.

## SARS-CoV-2 inhibition assay: Viral replication

For the evaluation of SARS-CoV-2 replication, the number of SARS-CoV-2 copies was measured by reverse transcription quantitative polymerase chain reaction (RT-qPCR). Individual VHHs were pre-incubated with Vero E6 cells (ten specified concentrations) 2 hours prior infection. Each plate included PBS (2 µL, negative control) (or PBS 0.25 % DMSO) and remdesivir (25 µM; SelleckChem) as a positive control. After the pre-incubation time, individual VHHs and remdesivir were removed, and the cells were exposed to the different variants of SARS-CoV-2 (at a pre-established multiplicity of infection: MOI of 0.07 (Delta, XBB.1.16.1), 0.27 (BQ.1.1, BA.1), and 0.33 (XBB.1.5) PFU/Vero E6 cells). After a one-hour adsorption at 37 ° C, the supernatant was aspirated and replaced with 2% FBS/DMEM media containing the respective VHHs or remdesivir at the indicated concentrations. The cells were then incubated at 37 ° C for 48 (B.1.617.2; BQ.1.1; XBB.1.16.1; BA.1) or 66 hours (XBB.1.5). The MOI and incubation time were defined previously to obtain the same rate of viral production after infection. Supernatants were collected and heat inactivated at 80° C for 20 minutes. The Luna Universal One-Step RT-qPCR Kit (New England Biolabs) was used for the detection of genomic SARS-CoV-2 RNA through RT-qPCR using a thermocycler (QuantStudio 6 thermocycler; Applied Biosystems). Specific primers targeting the N-region viral gene: Forward: 5' TAATCAGA-CAAGGAACTGATTA Reverse: 5' CGAAGGTGTGACTTCCATG were used. Cycle threshold (Ct) were determined by the second derivative maximum method in the QuantStudio software v2.6.0. The quantity of viral genomes is expressed as Ct and was normalized against the Ct values of the negative (PBS) and positive controls (remdesivir 25 µM). Curve fits and IC50 values were obtained in Prism using the variable Hill slope model.

## Cell viability assay

Cell viability assays were performed in compound-treated cells using the CellTiter-Glo assay according to the manufacturer's instructions (Promega) and a Berthold Centro XS LB960 luminometer (MikroWin 2000). 3,000 cells/well of Vero E6 were seeded in white with clear bottom 384-well plates. The following day, compounds were added at concentrations indicated. PBS only and 10 µM camptotecin (Sigma-Aldrich) controls were added in each plate. After 48 h incubation, 10 µL of CellTiter Glo reagent was added in each well and the luminescence was recorded using a luminometer (Berthold Technologies) with 0.5 sec integration time. Raw data were normalized against appropriate negative (0 %) and positive controls (100 %) and are expressed in % of cytotoxicity.

## Crystallization and structure determination

Purified hACE2 was incubated overnight at 4 ° C with VHHs B07 and B10 at final concentrations of 23 µM, 129 µM, and 129 µM, respectively. Then, a SEC step was performed to isolate the complex peak from unbound fractions. The complex was purified to 7.9 mg/mL and used for an initial crystallization screening trials by flowing the procedure established at the Crystallography Core Facility of the Institut Pasteur[69]. Briefly, crystallization drops were generated by mixing equal volumes (200 nL) of the protein complex and reservoir solution in the format of 96 Greiner plates, using a Mosquito robot, and monitored for crystal growth by a Rock-Imager. Crystals grew at 18° C in 10% (w/v) 2-propanol, 20% (w/v) Polyethylene glycol 4000, and 0.1 M HEPES pH 7.5. For cryoprotection, the crystals were soaked in crystallization solution supplemented with 20% glycerol before flash-frozen under

liquid nitrogen. The crystals obtained were tested at beamlines PX1 and PX2 at SOLEIL synchrotron (Saint Aubin, France). The best dataset was collected at beamline PX2-A on a EIGER X 9 M area detector at a resolution of 2.7 Å, and was indexed, integrated, scaled, and merged using XDS[70] and AIMLESS[71]. The high-resolution limit was determined using CC1/2-based cutoffs of 30%[72]. The structures of molecules were then determined by molecular replacement with PHASER[73] using the search models of hACE2 from PDB code 1R42 and alpha-fold models for VHH B07 and VHH B10[74]. The asymmetric unit (au) contained 3 complexes of hACE2-VHH B07-VHH B10. After reconstruction of subdomains of hACE2 and of CDRs of VHHs, the models were alternatively manually corrected using COOT[75] and refined using non-crystallographic symmetry and TLS parameterization (Translation/Libration/Screw)[76] with BUSTER-TNT[77]. The final R and free R factors are 21.9% and 25.4%, respectively. The final refined models were analyzed with MolProbity[78]. The data collection and refinement statistics are listed in Supplementary Table 2.

## Structural data analysis

The figures of the structures were prepared using the PyMOL Molecular Graphics System, version 2.0.0 (Schrödinger LLC) (http://pymol.sourceforge.net). The polar interactions (hydrogen bonds and salt bridges), the contact residues (van der Waals) and the accessible (ASA) and buried (BSA) surface areas were assessed using 'Protein interfaces, surfaces and assemblies' service PISA at the European Bioinformatics Institute[79] (http://www.ebi.ac.uk/pdbe/prot_int/pistart.html). The intermolecular interactions were computed using a maximal cut-off distance of 3.9 Å for polar contacts (PISA distance cut-off) and 4.75 Å for van der Waals contacts (PyMol distance cut-off).

For the final analysis of contacts between molecules (au containing 3 complexes), we used the chain A for hACE2, chain D for VHH B10 and chain E for VHH B07 (chains D and E have lower B-factors compared to other chains of VHHs, indicative of a more orderly disposition of atoms within the crystal). However, for all the complexes the sites of interactions between hACE2 and VHHs are ordered.

Electrostatic surfaces were computed using the PDB2PQR[80] and APBS[81] calculations software. Sequence alignments were performed using clustal W omega[82] and plotted using ESPript server[83]. The accession numbers of ACE2 sequences are: Human (UniProt ACE2_HUMAN); Hamster (UniProt A0A1U7QTA1_MESAU); Mouse (UniProt ACE2_MOUSE). The VHH sequences were analysed by Abysis (http://www.abysis.org/abysis) and IMGT[84] (https://www.imgt.org) websites for mapping CDR/framework regions according to Kabat[85] and IMGT conventions, respectively. The analysis of the putative germline and somatic maturation events was done with the IMGT website.

## Immunofluorescence imaging of primary human nasal epithelium cells and K18-hACE2 mouse tissues

MucilAir™, reconstructed human nasal epithelial cells (hNECs), previously differentiated for 4 weeks, were cultured in 700 μL MucilAir™ media on the basal side of the air/liquid interface (ALI) cultures as described[86]. For imaging, cells were fixed on the apical and basal sides with 4% PFA for 15 minutes. Cells were stained O/N at 4° C with anti-hACE2 VHH-B07-Fc at 6 μg/mL revealed with a Goat anti-Human Alexa Fluor-594 secondary Ab (Jackson Immuno Research, #109-587-003, 1:200), added with Phalloidin Atto-647 (Sigma-Aldrich 65906, 1:400) (F-actin staining), and DAPI (Thermo Fisher, 1 mg/mL) (Fig. 6d) or they were co-stained with anti-hACE2 VHH B07-Fc and rat anti-alpha-tubulin (ThermoFisher, # MA1-80017, 1:200) revealed with a Goat anti-Human Alexa Fluor-594 and a Donkey anti-Rat Alexa Fluor 488 (Thermo Fisher scientific, #A-21208, 1:200) secondary antibodies (Supplementary Fig. 9c).

For immunofluorescence experiments on whole-mount preparations of mouse organs, following euthanasia, samples of nasal turbinates and lungs of K18-hACE2 mice were collected and fixed in phosphate-buffered saline (PBS) containing 4% paraformaldehyde for 24 h at 4° C. Then, nasal respiratory epithelium, olfactory epithelium and lung bronchioli were finely excised and post-fixed in PBS containing 4% paraformaldehyde for 1 hour, rinsed in PBS and immersed in blocking buffer (20% normal goat serum (NGS) and 0.3% Triton X-100 in PBS) for 1 hour at room temperature, and then incubated overnight at 4 ° C in primary antibodies (in PBS containing 2% bovine serum albumin (BSA)): Mouse anti-SARS-CoV-2 nucleoprotein (N) antibody (Genetex GTX36802, 1:100) and Rat anti-alpha-tubulin (1:200). After rinsing in PBS, samples were incubated for 2 hours at room temperature in PBS-2% BSA with the appropriate secondary antibodies: Goat anti-mouse Alexa-Fluor 555 (Abcam; ab150114, 1:200); Donkey anti-Rat Alexa-Fluor 488, Phalloidin Atto 647 (Sigma-Aldrich) (F-actin staining) and DAPI (Thermo Fisher Scientific, 1 mg/mL) were added in secondary antibodies mix. Fluorescent immunolabellings were imaged with a LSM-700 confocal microscope and ZEN software 2012 (Zeiss) and analyzed using ImageJ v2.16.0 (Fiji) software as described[37,86].

## SARS-CoV-2 infection and treatment in K18-hACE2 mouse model

B6.Cg-Tg(K18-ACE2)2Prlmn/J mice (stock #034860) were imported from The Jackson Laboratory and bred at the Institut Pasteur under strict specific pathogen-free conditions. Infection studies were performed on 10–13 week-old male and female mice, in animal biosafety level 3 (BSL-3) facilities at the Institut Pasteur. All animals were handled in strict accordance with good animal practice. Mice were housed in groups of up to 5 individuals, in negative pressure isolators, under 12 h:12 h light-dark cycle, at 24 +/-1.5 ° C temperature and 50-70% relative humidity. They were fed standard chow ad libitum, with autoclaved water. Animal work was approved by the Animal Experimentation Ethics Committee (CETEA 89) of the Institut Pasteur (project dap 210050), and authorized by the French Ministry of Research (under project 31816) before the experiments were initiated. Twenty-four hours before infection, anesthetized (ketamine/xylazine) mice were administered intra-nasally (i.n.) with either 0.9% NaCl (40 μL), control VHH (7 mg/kg or 5 mg/kg in 20-30 μL) or B07-Fc (7 mg/kg in 50 μL or 5 mg/kg in 30-50 μL). On the next day, they were again anesthetized (ketamine/xylazine) and inoculated i.n. with $1 \times 10^5$ PFU of SARS-CoV-2 XBB.1.5 (hCoV-19/France/PDL-IPP58867/2022; 10 μL/nostril) or $1 \times 10^4$ PFU of SARS-CoV-2 D614G (hCoV-19/France/GE1973/2020; GISAID ID: EPI_ISL_414631; 15 μL/nostril). Clinical signs of disease (ruffled fur, hunched posture, reduced mobility and breathing difficulties) and weight loss were monitored everyday (except weekend). Each of the four criteria received a 0–2 score and was added to a global score. Mice were euthanized when reaching 25% body weight loss or a global score of 8 or euthanized at specific time points for tissue collection. On day 3 or 6 post-infection, mice were euthanized by cervical dislocation. The trachea and upper lung lobes were fixed by submersion in 10% phosphate buffered formalin for 24-36 hours prior to removal from the BSL3 for processing and transfer in 70% ethanol. Lower lung lobes (and nasal turbinates in a subset of mice) were dissected and frozen at -80° C. Samples were homogenized in 500 μL of PBS using lysing matrix M (MP Biomedical) and a MP Biomedical FastPrep 24 Tissue Homogenizer. For lung viral load measurement, genomic viral RNA was extracted using an extraction robot IDEAL-32 (IDsolutions) and the NucleoMag Pathogen extraction kit (Macherey Nagel). Viral RNA quantification was performed by quantitative reverse transcription PCR (RT-qPCR) using the IP4 set of primers and probe (nCoV_IP4-14059 Fw GGTAACTGGTATGATTTCG and nCoV_IP4-14146 Rv CTGGTCAAGGTTAATATAGG giving a 107 bp product, and nCoV_IP4-14084 Probe(+)TCATACAAACCACGCCAGG [5']Hex [3']BHQ-1) and the Luna Universal Probe One-Step RT-qPCR Kit (NEB). Serial dilutions of a titrated viral stock were analyzed simultaneously to express viral loads as PFU equivalents (eqPFU) per gram of tissue.

For plaque assay, 10-fold serial dilutions of samples in DMEM were added onto VeroE6 monolayers in 24 well plates. After one-hour

incubation at 37° C, the inoculum was replaced with equivalent volume of 5% FBS DMEM and 2% Carboxymethylcellulose. Three days later, cells were fixed with 4% formaldehyde, followed by staining with 1% crystal violet to visualize the plaques.

## Measurement of VHH concentration in lungs homogenate by ELISA

sACE2 (2 µg/mL) were coated on polystyrene plate (Thermo Fisher Scientific) over night at 4° C. After washing with PBS/Tween 0.1%, plates were incubated with B07-Fc (from 0.07 to 10 ng/mL) or homogenate of lower lung lobes in PBS 0.5% Gelatin, 0.1% Tween-20 (PGT solution) for 90 min at 37° C. A negative control (sACE2 alone) was prepared under the same conditions. Then, plates were incubated at 37° C with Peroxydase Affinipure rabbit anti Human IgG (Jackson ImmunoResearch, AB_2339647, 1/1000) in PGT for 45 min and treated with orthophenylenediamine solution containing 0.1% hydrogen peroxide (H2O2) for 5 min. The reaction was stopped by adding chlorohydric acid 3 M (HCl) and the absorbance of the wells was measured at 492nm in a spectrophotometer-microplate reader (Sunrise, Tecan, software XFLUOR4 v4.51).

## SARS-CoV-2 infection and treatment in Syrian golden hamster

The animals used were male Syrian golden hamsters (*Mesocricetus auratus*, strain RjHan:AURA), 6 week of age (weight between 70−90 grams), obtained from Janvier Laboratories. Golden hamsters were housed and manipulated in class III safety cabinets in the Institut Pasteur animal facilities accredited by the French Ministry of Agriculture. Animal work was approved by the Animal Experimentation Ethics Committee (CETEA) of the Institut Pasteur (project dap 210011) and authorized by the French legislation (project #21045) in compliance with the European Communities Council Directives (2010/63/UE, French Law 2013-118, February 6, 2013) and according to the regulations of Institut Pasteur Animal Care Committees before the experiments were initiated. Hamster were housed in groups of up to 4 individuals, in negative pressure isolators, under 14 h:10 h light-dark cycle, at 22 + /-2° C temperature and 55 % ( + /- 10 %) relative humidity. Anesthetized animals received intranasal inoculation (5 mg/kg) of VHH-B07-Fc or VHH-ctl-Fc. Twenty-four hours after inoculation, hamsters were infected intranasally (i.n.) with $10^4$ PFU of SARS-CoV-2 XBB.1.16.1 as previously described[67].

All hamsters were followed up daily. At day 3 post-infection, animals were euthanized with an excess of anesthetics (ketamine, xylazine and laocaine) (AVMA Guidelines 2020), and samples collected. 100 mg lung fragments were ground in Lysing Matrix X tubes (MP Biomedicals) using a grinder at a speed of 6.5 m/s for 30 seconds and centrifuged at 2,000 g for 10 minutes (4° C) to clarify the supernatant.

For the evaluation of the number of SARS-CoV-2 copies in tissues by RT-qPCR, 100 µL of each grinding solution was mixed with 500 µL of Trizol to deactivate virulence. The total RNA was then extracted using the Direct-zol RNA MiniPrep Kit (Zymo Research) and quantified. The presence of genomic SARS-CoV-2 RNA in these samples was evaluated by one-step RT-qPCR in a final volume of 5 µL per reaction in 384-well PCR plates using a thermocycler (QuantStudio 6 thermocycler v2.6.0; Applied Biosystems). The primers used targeted N-region viral gene: Forward: 5' TAATCAGACAAGGAACTGATTA Reverse: 5' CGAAGGTGTGACTTCCATG. Viral load quantification (expressed as RNA copy number/g lung) was assessed by linear regression using a standard curve of seven known quantities of RNA transcripts containing the N-region viral gene. All collected data were plotted and analyzed using GraphPad Prism software. Mann-Whitney tests were performed for statistical analysis.

## Statistical analysis

Figures were generated using GraphPad Prism 10 (GraphPad software). Statistical analysis was conducted using GraphPad Prism 10. Statistical significance between different groups was calculated using the tests indicated in each figure legend.

## Reporting summary

Further information on research design is available in the Nature Portfolio Reporting Summary linked to this article.

## Data availability

All data supporting the results of this study can be found in this article, the Supplementary Information, and the Source data file. Coordinates and structure factors have been deposited in the Protein Data Bank under the accession code 9R19 (https://www.rcsb.org/structure/9R19). The unique materials used in this study are readily available from the authors upon request (Materials Transfer Agreement with the Institut Paseur) for academic research purposes. Source data are provided with this paper.

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

## Acknowledgements

We are grateful to Julian Buchrieser (Virus and Immunity unit, Institut Pasteur) and Erika Cecon (Institut Cochin, Paris, France) for the gifts of vectors expressing ACE2 and Hugo Mouquet (Laboratory of Humoral Immunology, Institut Pasteur, Paris, France) for anti-spike mAb antibody. We acknowledge the Molecular Biophysics Platform (Institut Pasteur, Paris) for soluble proteins quality control, in particular Bertrand Raynal, Sandrine Rosario, and Sébastien Brule. We thank Gaelle Chauveau and Catherine Vivier of the Antibody Engineering platform (Institut Pasteur) and Mireille Nowakowski of the Production and Purification of recombinant Proteins platform (Institut Pasteur) for their help with the production of VHHs. We acknowledge the National Reference Centre for Respiratory Viruses hosted by Institut Pasteur (Paris, France) for virus collections. We thank the staff of the Animal Facility (C2RA) for the breeding and care of the K18-hACE2 mice. We acknowledge SOLEIL (Saint-Aubin, France) for provision of synchrotron radiation facilities, and we thank the staff of beamlines PROXIMA-1 and PROXIMA-2A for assistance. The authors are grateful to the staff of the crystallography platform at the Institut Pasteur for robot-driven crystallization screenings. We thank Eduard Baquero Salazar of the NanoImaging platform (Institut Pasteur) for helpful discussion on the sACE2-B07-B10 structure. We acknowledge the members of the Direction des Applications de la Recherche et des Relations Industrielles (DARRI, Institut Pasteur, Paris) involved in the project, particularly Corinne Sarrazin, Serge Pauillac, and Michel Perez from the patent office. We thank all the members of the Dynamics of Host-Pathogen Interactions Unit (Institut Pasteur) for help with experiments and discussions, particularly Geneviève Janvier. We thank Françoise Porrot and Florence Guivel-Benhassine (Virus and Immunity unit, Institut Pasteur, Paris) for virus production. Institut Pasteur "Programme Federateur de Recherche" (PFR-6 SI-COV-2 project), Urgence COVID-19 Fundraising Campaign of Institut Pasteur (FAR, JE, OS, ABr). Institut Pasteur DARRI/CARNOT « Maturation Program » (PL, XM, FAR, ABr). The French Government's Investissement d'Avenir program, Laboratoire d'excellence "Integrative Biology of Emerging Infectious Diseases" (grant ANR-10-LABX-62-IBEID) (XM, FAR, JE, OS, ABr). Programme Fédérateur de Recherche SARS-CoV-2 & COVID-19, PFR-4 Long Covid (XM). Laboratoire d'excellence "Milieu Interieur" (JE). Fondation pour la Recherche Médicale (FRM) (OS). ANRS (OS). The Vaccine Research Institute (ANR-10-LABX-77) (OS) ANR / FRM Flash Covid PROTEO-SARS-CoV-2 (OS). ANR Coronamito (OS). HERA european funding (OS). Sanofi (OS). IDISCOVR (OS). ANRS-MIE (BIOVAR and PRI projects of the EMERGEN research program) (XM, FA). ANRS-MIE Project EMERGEN - ANRS 0149 - PRODEVAR and ANR-22-CE35-0004 BAT-CoV-ASIA (FAR). SB is the recipient of an MESR/Ecole Doctorale BioSPC, Université Paris Cité, fellowship; FS is the recipient of an Erasmus program. DP is supported by the Vaccine Research Institute. The Opera system was co-funded by Institut Pasteur and the Région île de France (DIM1Health). Funding sources are not involved in the study design, data acquisition, data analysis, data interpretation or manuscript writing.

## Author contributions

Conceptualization: ABr Proteins productions: IF, AA, ED, PL, GA Alpaca immunization: PL, GA In vitro assays: SB, IS, FS, ABo, JC, JT-R, PL, GA, ABr X-ray crystallography: M-CV, IF, AA, AH, FAR In vivo assays: LC, EG, SG Viral strains collection: XM, FA, OS Imaging: VM, DP Supervision: AH, SP, XM, FA, PL, OS, FAR, JE, ABr Writing—original draft: ABr wrote the manuscript with contributions from all authors.

## Competing interests

SB, IF, IS, FAR, PL, GA, OS, ABr are designated as inventors in patent application No. WO 2025/056771 A1 filed by Institut Pasteur. The patent application covers the aspects of anti-ACE2 VHH B07, B09, B10 described in the manuscript. The other authors declare no competing interests.

## Additional information

[1]Institut Pasteur, Université Paris Cité, Dynamics of Host-Pathogen Interactions Unit, CNRS UMR3691, F-75015 Paris, France. [2]Institut Pasteur, Université Paris Cité, Structural Virology Unit, CNRS UMR3569, F-75015 Paris, France. [3]Institut Pasteur, Université Paris Cité, Mouse Genetics Laboratory, F-75015 Paris, France. [4]Institut Pasteur, Université Paris Cité, Mouse Genetics, Immunity and Infections Laboratory, F-75015 Paris, France. [5]Institut Pasteur, Université Paris Cité, Virus and Immunity Unit, CNRS UMR3569, F-75015 Paris, France. [6]Institut Pasteur, Université Paris Cité, Chemogenomic and Biological Screening Core Facility, C2RT, CNRS UMR3523, F-75015 Paris, France. [7]Institut Pasteur, Université Paris Cité, Pathogenesis of Vascular Infections Unit, INSERM U1225, F-75015 Paris, France. [8]Institut Pasteur, Université Paris Cité, Production and Purification of recombinant Proteins Platform, CNRS UMR3528, F-75015 Paris, France. [9]Institut Pasteur, Université Paris Cité, Crystallography Platform-C2RT, CNRS UMR3528, F-75015 Paris, France. [10]Vaccine Research Institute, Créteil, France. [11]Institut Pasteur, Université Paris Cité, Antibody Engineering Platform, CNRS UMR3528, F-75015 Paris, France. [12]These authors contributed equally: Laurine Conquet, Isabelle Staropoli. [13]These authors jointly supervised this work: Gabriel Ayme, Olivier Schwartz, Felix A. Rey, Jost Enninga. ✉e-mail: anne.brelot@pasteur.fr

