## [Transparent Peer Review file · Nature Communications]

Targeting ACE2 with a camelid antibody inhibits SARS-CoV-2 binding and has protective effects in vivo

Corresponding Author: Dr Anne BreLOT

Version 1:

Reviewer comments:

Reviewer #1

(Remarks to the Author)

Summary:

The manuscript developed a pan-SARS-CoV-2 neutralizing antibody B07-Fc for intranasal administration. The paper presents a methodical and well characterization of the VHH starting from its production, panning selection, and detailed in vitro binding assays, including binding affinity studies and functional competition. The authors also effectively demonstrate that these VHH do not interfere with ACE2 enzymatic activity, which is crucial for its development as a therapeutic candidate that will not influence its biological role. The authors have tested B07 across a range of SARS-CoV-2 variants, highlighting the potential of B07 to be effective against a broad spectrum of viral mutations. The author shows that the fusion of VHH candidate B07 to an Fc domain works effectively, enhancing the therapeutic potential by improving binding affinity and functional efficacy. Finally, the authors evaluated the neutralization efficacy of B07-Fc in the K18-hACE2 mouse model and showed that B07-Fc protect against SAR-CoV-2 transmission to lung in three days post viral challenge. However, a long-term mouse study including multiple additional study are required to conclude the antibody efficacy and stability. Since intranasal administration is most effective against initial viral infection, the author should emphasize more on the significance of intranasal administration compared to conventional approach. In conclusion, while the manuscript presents promising findings, significant issues weaken the study's overall impact. The lack of robust statistical analyses and insufficient in vivo efficacy data undermine the comprehensiveness of the study.

Major comments:

1. In Fig 2B, the number of data points of B09 and B10 groups (only 2 samples) are different than ctrl and B07 samples (5 samples).
2. Authors should show statistical analyses (p-values confidence intervals (CIs) for VHH IC50 values, and R² values for dose-response curves) on the figures.
3. The flow data in Fig 2C is inconsistent with the quantitative data in Fig 2B. From flow, geo mean of B09 binding to IGROV-1 show a higher shift compared to B07 while the quantitative data show the other way around.
4. Fig 3B, C, Fig 5E lacks mock control to set the 100% enzymatic fluorescence baseline.
5. In Fig 3D, Author should briefly explain why fluorophore (ACE2 enzymatic activity) increased.
6. Fig 5G, to demonstrate the inhibition of SARS-Cov-2 infection by selected VHHs, an assay that directly shows the reduction of infected cells may be more relevant. For example, use GFP-labeled SARS-CoV-2 to infect the VHHs pre-incubated cells. Adding spike protein immunostaining in the S-Fuse assay is also recommended.
7. In Fig 6B, since mouse and hamster ACE2 share identical sequence, why B07 bind to hamster and mouse ACE2 different affinity?
8. In Fig 6B, structure or docking prediction of B07 VHH interaction with human ACE2, hamster ACE2, mouse ACE2 is highly recommended.
9. The animal study in Fig 7 is not sufficient to support the conclusion that "Fc-conjugated B07 dimers protect against Omicron infection in vivo". In Fig 7A, the study only observed infected mice for up to 3 days. The study should include a long-term evaluation (up to 14-21 days). This also help to assess B07-Fc half-life and strengthen the claim of B07-Fc protection against viral infection.
10. In Fig 7B, as previously mentioned, the manuscript does not provide statistical comparisons between the control (ctl-Fc) and treated (B07-Fc) groups. Only 6 out of 10 mice showed a reduction in viral load is not convincing to demonstrate the B07-Fc protection. Author explanation of the variability is due to heterogeneity. However, if the variability is due to the sub-optimal distribution of B07-Fc in the lungs, the authors should propose or implement methods to improve this distribution,

ensuring consistency across the treated animals. If the variability is potentially due to VHH degradation or aggregation, it is important to conduct tests to assess these factors.

11. Compared to qPCR-based titration assay shown in Fig 7B, a conventional plaque assay may be more relevant to actual viral titer.

12. Fig 7C, it would be more direct to see lung infections if the author shows immunostaining of the SARS-CoV-2 virion particle.

13. Fig 7. Infection of K18-hACE2 mice with SARS-CoV-2 can lead to lethality. Therefore, to better illustrate intranasal B07-Fc protection from lethal challenge, a mouse survival curve should be included. It would be also interesting to see the immune profile of the challenged mice.

14. Fig 7. Since the neutralizing antibodies were produced using the E.coli system, it seems necessary to supplement experiments regarding the removal of endotoxins such as LPS. For example, in the case of in vitro modeling, the study was conducted based on HEK293 cells, which have minimal expression of toll-like receptors, so there might be insufficient validation of potential side effects. Furthermore, in the animal model, there are only results for intranasal administration. Because intranasal only protect viral early intranasal infection, and stability of B07-Fc was not assessed, it seems necessary to add cross-validation for other commonly used administration methods such as IP or IV injection, as well as data on the degradation of neutralizing antibodies within the animal body.

Minor comments:

1. In introduction section, pan-neutralizing antibody can be also achieved by directly targeting the RBD of the S protein from SARS-CoV-2 other than anti-hACE2 antibody that the author proposed. Both strategies have its own advantages and disadvantages. The manuscript should include this explanation and highlight the study novelty.

2. In Supplementary Figure 2, the incubation time with B07-Fc is not specified.

3. Supplement Fig 3B, the fluorescence voltage should be adjusted to place the control population more towards center.

4. Intranasal administration dosages in hamsters (Supplement Fig 4, 5mg/kg) and mice (Fig 7, 7mg/kg) are different, which may count for the VHH neutralization efficacy difference.

5. Authors should explain the rationale of using clone B07 instead of B09.

6. Standardize font use. The font used in the table in Fig 4A, legend in Fig 5 A, table in Fig 5F and G etc. all use different font than others.

7. Fig 7C fluorescence have similar color, which make the graph hard to read. Use of distinct wavelength fluorophores is highly recommended.

8. Authors need to refine figure titles to better conclude the overall context of figures.

Reviewer #2

(Remarks to the Author)

In this manuscript, Blachier et al. present a study on a novel approach to prevent SARS-CoV-2 infection using single-domain antibodies (VHHs) that target the ACE2 receptor, rather than the spike protein. This strategy is particularly relevant given that some SARS-CoV-2 variants have developed resistance to vaccines and therapies targeting the spike protein. The authors demonstrate that targeting ACE2, the critical entry receptor for the virus, can prevent infection of multiple SARS-CoV-2 variants, including recent Omicron subvariants.

The authors identified and characterized three VHHs (B07, B09, and B10) isolated from an alpaca immunized with ACE2. These VHHs bind ACE2 with high affinity. Two VHHs (B07 and B09) prevent entry of multiple SARS-CoV-2 variants through a competitive mechanism that prevent virus attachment to host cells. In contrast, VHH B10, does not inhibit viral infection, underscoring the importance of epitope specificity in preventing virus attachment. None of the characterized VHHs affect the catalytic activity of ACE2, which is essential for maintaining normal physiological function.

The authors create dimers by fusing the VHHs to an Fc domain, resulting in improved binding and prevention of viral entry.

Overall, the study is well written and well conducted and highlights an interesting concept to prevent infections.

I am surprised the authors call the mechanism neutralization even though the VHHs do not target the virus, but the host. The VHHs prevent entry and infection, but they do not per se neutralize. I recommend adjusting the wording to describe the function of the VHHs more accurately.

Otherwise, I recommend the manuscript for publication.

Reviewer #3

(Remarks to the Author)

This manuscript reports the work of Blachier, et al to identify a single-domain antibody (VHH) derived from an alpaca as a potential therapeutic antibody against SARS-CoV-2 (SARS2). ACE2 is an essential protein for the entry of SARS2 and is considered to be the receptor that triggers the viral spike protein (S) to undergo conformational changes that result in the fusion of the viral membrane with the target cell membrane. This is the penultimate step in virus entry (uncoating being the last) and is essential for SARS2 virus infection. Thus, inhibition of ACE2 binding by S is a prominent target of host immune responses, SARS2 vaccines and SARS2 therapeutic antibodies. Because of host immune responses to previous infection and vaccination, SARS2 has undergone significant evolutionary changes in its spike protein, leading to escape from both existing immunity and therapeutic antibodies targeting S. To potentially mitigate against this, the authors chose to target the host cellular protein ACE2 because the number of allelic differences are small and the much smaller rate of evolutionary change in human ACE2 make it a stable therapeutic target.

Single-domain antibodies or nanobody are purported to have advantages over “typical” IgG antibodies (mAbs) that have

both heavy and light chains, with most of the theoretical benefits relating to size and their ability to have long CDR3 regions. However, there is a relative lack of data on their clinical use and benefits over classical mAbs and their small size means they are likely to have pharmacokinetic properties similar to that of antigen binding fragments (Fabs) that are derived from mAbs (i.e. have very short half-lives). In addition, they often lack the benefit of the avidity provided by the multivalency of traditional IgG mAbs and the ability to induce antibody mediated effector functions that require an Fc domain. To overcome these issues, VHH's are often fused to Fc domains to create bivalency, add effector functions and increase their half-lives.

In this report, the authors identify 3 VHH (B07, B09, B10) that target the ACE2 binding site that is bound by S and perform a limited evaluation of their epitopes and ability to block the fusion step in virus entry for multiple variants of SARS2. They perform a more in-depth analysis of their leading candidate B07 and fuse it to an Fc domain to provide enhanced avidity and make it more IgG like. They subsequently use a humanized ACE2 mouse model to show that there is likely a benefit when given as a prophylactic and intranasal therapy. Data in hamsters was not as robust, but they show that this may be due to a significant difference in the ability of B07 to bind to hamster ACE2.

Overall, the approaches are sound, the experiments are done well and the results generally support the conclusions and claims that are made. There are a few issues that need to be addressed and some conclusions that need to be tempered.

Concerns to be addressed:

- 1) Line 88. VHH are not fragments as they are complete naturally smaller proteins.
- 2) Line 90. While it is true that VHH can be produced in prokaryotic systems, the use of an Fc domain abrogates this benefit.
- 3) Line 93. "Therefore, they can be delivered to the lung by inhalation, which can confer significant advantage for SARS-CoV-2 treatment." It is not clear how the properties described in the preceding sentences means that VHH have properties that allow it to be delivered to the lungs by inhalation where other therapies cannot -- especially if the VHH has been fused to an Fc. Traditional antibodies have also been delivered to the lungs by inhalation and the intranasal route.
- 4) Line 111. "constructed a cDNA library containing about 8×10^7 different VHHs sequences. The method only discusses PBMCs but not how many B cells went into the library and the method estimates a library size of 8×10^7 cfu (colony forming units?) and the text says VHHs sequences. Is it cfu or sequences? If the later, how was that number estimated. Was the whole library sequenced? And if sequenced, how many unique lineages were represented in 8×10^7 sequences. If it is cfu, then the text needs to be corrected because likely that there are many copies of the same VHH cfu library. Either way, how many input cells were used to create that number of VHHs.
- 5) Line 113-115. "192 individual clones were assayed by ELISA". If it is known if these were unique sequences in the clones, please provide the number of unique as it will help put the small number of positive clones in context.
- 6) Line 116-117. If B07 and B10 differ only by 6 CDR amino acids are they from the same lineage? If so, please indicate. Similarly, please provide information on which CDRs and if there are framework mutations.
- 7) Please provide supplementary figure showing the sequences for each VHH and include an alignment of B07 and B09 showing their differences
- 8) Lines 119-133/Figure 1. BLI measurements of VHH affinities to soluble ACE2 (sACE2).
 - a. Please comment if sACE2 was monomeric or dimeric form. It appears to be dimeric and if so, the affinities are likely apparent KD because the VHH is bound to the sensor chip. The model used for curve fitting does not match the raw curves well and this could be indicate that sACE2 is a dimer and/or that a 1:1 model may not be the best to use. Please justify the use of a 1:1 fitting model. Alternatively, it would be best to put the sACE2 on the sensor tip and use the VHH as the analyte in solution.
 - b. Please convert concentrations from ug/ml to nM for the concentrations of sACE2 used.
 - c. Please repeat BLI measurements for B07 and B08 to include a concentration range of sACE2 where the upper value is at least 10-fold more than the measured IC50. This is the standard recommendation to ensure an accurate measurement.
- 9) Lines 124-125. The affinity for B07 and B08 are poor (~350 nM). High affinity is generally single digit nM affinity or lower.
- 10) Line 135 and later. It would be helpful to the reader to indicate human ACE2 where appropriate because mouse, hamster and other species ACE2 are used in different locations. For example, endogenous ACE2 on Vero cells (Line 138) is a non-human primate.
- 11) Line 140. Does the control VHH also have a myc-tag? If not, assay will need to be repeated using a VHH control with a myc-tag so it matches the experimental VHHs and their detection. If so, please modify text to state that.
- 12) Figure 2 ACE2 surface binding. Please provide a positive ACE2 binding control. These are commercially available to purchase. In addition, please provide a control that tests binding of SARS2 spike or RBD. These will help show the expected binding ranges based on the ACE2 expression levels for the VHHs.
- 13) Line 150. In order to determine if 10 uM is an appropriate amount of VHH to use for an off-target enzymatic inhibition assay, the manuscript should provide a rationale for choosing 10 uM. The value should be chosen based on an expected Cmax and steady state concentration of the VHH. They could choose the value based on human or animal studies of VHH therapeutics. Offhand 10 uM seems too low.
- 14) Inhibition assay. The manuscript would be improved by providing the reader with some basic information about the in vitro and in cellulo ACE2 enzymatic cleavage assays, substrate and positive control inhibitors in the text. Some of the information is present but it needs to be rephrased in a way that brings it all together and adds one or two small missing bits.
- 15) "Neutralization assays" Lines 160 and below. In neutralization assay, virus particles are incubated with increasing amounts of antibodies or VHH prior to being added to cells. Following a set period, the number of cells that are infected is measured. The differences in number of cells infected when antibody is present, and no antibody is defined as the neutralization. SARS2 entry is a multistep process that includes cellular attachment, receptor binding, fusion and uncoating. In addition, there is also an endosomal entry pathway involving proteolytic cleavage and a cell surface pathway utilizing TMPRSS2 (Nature Reviews Molecular Cell Biology volume 23, pages3–20 (2022)). Thus, each of these steps can be targeted by an antibody and can influence an antibodies activity. Thus, fusion assays are a proxy for neutralization activity and while they can be used to screen for neutralization, the term neutralize or neutralization should not be used as they only

model one aspect of entry. Please modify the manuscript and figures labels to reflect the assays as a fusion inhibition assay. 16) While it is reasonable to screen for activity of multiple variants using the fusion assay, it is possible that the VHHs might fall off of ACE2 during transport into endosomes or at the low pH conditions found in endosomes. The authors should first perform a traditional virus or pseudovirus neutralization assay using at least one variant to demonstrate that the neutralization IC50 for the VHHs are similar to that seen in the fusion assay. This would provide the justification to use a fusion assay as a proxy for neutralization of other variants.

17) Since it is likely that the VHH will see low pH, the manuscript would be improved by determining KD values for B07, B09 and B-7-Fc to sACE2 at a pH that would be found in endosomes (e.g. pH 5.5).

18) Line 167-169: There is quite a broad range of IC50's seen. Please discuss. Does this correlate with the affinity of ACE2 for the variant RBD's? Suggest adding a correlation analysis to the data set and add to manuscript figure or supplement.

19) Line 190. "to stabilize the VHHs and increase their efficacy."

a. What is meant by stability and how is being in an IgG form going to help with stability? Up until this point stability of the VHHs has not been mentioned. Please speak to what the stability issue is and how IgG1 fusion solves this. Alternatively, remove the word stability.

b. Since animal efficacy of the VHH vs the Fc-VHH forms is not studied, efficacy improvement is evaluated in this report. I would recommend changing to potency because that is being measured and is shown to be improved. Alternatively, efficacy can be removed.

20) Lines 205-209. T

a. For the comparisons of IC50s in the fusion assay it is confusing to have a dash between the conversion of ug/ml to molar because it seems that the ug/ml is the VHH range and the nM is the VHH-Fc range. I would recommend removing the ug/ml range and adding in the nM converted B07 BHH result in its place.

b. Similarly, I would show the number for B07-Fc and for the VHH version of B07 as nM.

21) Binding efficacy Section (Lines 214-233). The investigations of allelic polymorphism with the vicinity of the S binding site of ACE2 is a good experiment. But the best experiment would be to obtain a more accurate model of the location of B07 binding to determine which residues to test. A negative stain electron microscopy 3D reconstruction (NSEM-3D), deuterium exchange or full structure, would be able to provide a better footprint and provide mechanistic insights. NSEM-3D is relatively quick to get and when coupled with fitting models of ACE2 and a VHH could provide insights into the mechanism of action and candidate binding residues to test in ACE2. For example, comparing the 3D model with Spike:ACE2 models could inform whether competition is direct or indirect (i.e., does B07 block ACE2 by binding directly to S binding sites or is it a steric blockade). I think it is reasonable to obtain NSEM-3D.

22) Line 221. "correctly detected" is subjective. Please change this to something more objective or quantitative.

23) To better support claims of breadth and potency across time, more contemporary spikes should be evaluated in the fusion assay (assuming it is shown to be equivalent to neut) or neutralization assay.

24) Discussion:

a. There is a lot of emphasis on VHH advantages when the deliverable here is VHH-Fc which loses most of the advantages relative to a native IgG mAb. Limitations of both approaches should be addressed in the discussion.

b. The report is not the first to target ACE2. Please place this research in the context of previously reported work and how this effort is better and/or overcomes limitations of previous reports.

c. Line 277. "Resistance would require a change in the way S binds to the ACE2/VHH complex, likely incompatible with the fusogenic activity of the protein". This is not necessarily the case, as the viral spike could increase its affinity to ACE2 and thus outcompete the ability of the VHH to bind to ACE2. It might do this by making it off-rate much slower than that of the VHH.

d. Line 309. "This would lead to synergistic effects." To strong, suggest changing would to may.

e. Line 320: "high efficacy" is used when discussing affinity. Change efficacy to affinity.

25) Figure 5 legend: Moi and hours of the assay readout are not the same between viruses for replication-based neutralization assays. This can cause the observed differences in IC50 values being reported between variants and B07-Fc. A higher moi means more virus is present and therefore more spike proteins. The consequence of this is that more spike proteins are present on particles that need to be neutralized and the result is a shift to a worse IC50 relative to a lower moi virus. Similarly, increase assay readout time means that additional replication cycles are occurring and this results in the generation of more virus genome copies. Since the assay is measuring viruses by RT-qPCR, it would mean that there could be a falsely worse IC50 value for B07-Fc for viruses assayed at 66 hours vs those assayed at 48 hours. The authors need to provide adequate justification of varying the assay moi and time differences and provide evidence that the differences do not affect IC50 values for each variant. In the absence of that, they will need to report the comparisons done under the same conditions to avoid biases.

26) Figure 1B.

a. upper left panel.

i. It is confusing to show mock data as a schematic of the result as a panel. I recommend moving the panel to the supplement and replacing it with a cartoon description in the main text or supplement. Also, the title of the schematic panel should clearly indicate it is a schematic by saying "Schematic of" or something similar.

ii. Since ACE2 is bound to the sensor chip, it is confusing to have ACE2 in the title. I would just say VHH competition assay or VHH epitope mapping

b. What is being assessed is the ability of one VHH to block the binding of another VHH, so having ACE2 in the title is potentially confusing for a reader. I recommend changing other subpanel titles to "[VHH name] as competitor". For example, bottom left is "B09 as competitor".

27) Figure 2 and others. Please provide an example gating tree for all flow experiments in the supplement.

28) Figure 4 and elsewhere. Because the assay is a measure of fusion and not neutralization, please change "Neutralization (%)" to "Fusion inhibition (%)" or something similar.

29) Figure 4B/C/D and elsewhere. Since multiple variants are used in the paper, please indicate the variant spike name used in the graph label and/or figure legend text.

30) Figure 5A and elsewhere. Please ensure that BLI graphs have all the individual concentrations indicated (in molar) for each curve in the graph or legend (i.e., a range of concentrations or ug/ml is not acceptable).

31) Figure 5G.

a. Recommend moving the curve key from the XBB.1.16.1 panel to the far-left panel (i.e. Delta) since most readers read from left to right.

b. If I am correct, "Viral replication inhibition" percent would approach 100% at low concentrations and "(% of untreated cells)" would be 0% at low concentrations. The y-axis is labeled with both. I believe the latter is what is desired based on the data shown. Need to correct this.

32) Figure 6

a. Panel C (BLI).

i. Please provide the exact concentrations in nM for each curve in the panel or legend (as opposed to a range).

ii. In addition, the experiment needs to be repeated to determine if the 65 nM apparent KD value is correct because:

1. Only 3 sensorgrams are provided and 4 to 5 is the generally accepted minimum to have an accurate globally fitted KD value measured.

2. The concentration range does not encompass a concentration that is at least 10-fold higher than the determined KD.

b. Panel D and associated manuscript text

i. To support the conclusion that residues contained in region 30 to 41 of ACE2 of hamsters (especially Q34) are the molecular mechanism for a lower binding affinity of B07-Fc, they need to directly test the hypothesis. This can be done in a similar manner to what they have completed in panel A. One would expect the binding to be partially or fully restored by reverting the hamster residue to the human equivalent. Even better would be to also show that the inverse mutation leads to loss of activity. While the manuscript shows H35N, it is possible Q may be different. And could be tested. But to prove it is the dominant cause, the Q34 in hamster needs to be "reverted" to H34.

ii. Also, the authors pass up a ripe opportunity to further map the epitope of B07 by taking advantage of the mouse ACE2. In panel B, they show that B07 binding to mouse ACE2 is just rising around -4 log ug/mL which is 1-2 logs different from the hamster and 3-4 from the human. This suggests that there are further differences between hamster and mouse in the epitope. Panel D, shows that mouse and hamster have the exact same residues as each other in aa 30-41, suggesting that differences exist in the epitope outside this region. The authors could perform the same assay as in panel A but focused on aa residues differences found between hamster and mouse in the region analyzed for human allelic differences

33) Figure 7 and associated text

a. Please state in the method and legend if panels B and D are measuring genomic or subgenomic RNA.

b. Please add the white light only images in a supplement for each of the fluorescence images panel C

c. While differences in architecture of epithelium cells and bronchiole are interesting to see, it would be more informative to see immunohistochemical staining of the tissues for viral antigen compared to H&E sections of tissue. This more directly visual assess for the ability of the therapy to protect. Please provide this data

34) Supplemental figure 2. Add associated white light image for figure

Version 2:

Reviewer comments:

Reviewer #1

(Remarks to the Author)

This manuscript is strengthened with new data and structural insights, and the authors have address most of the reviewers' comments. Minor issues remain with grammar.

Minor comments:

1. Line 83, profile is misspelled as "profil".

2. Line 153, the correct expression should be "in cellulose" instead of "in cellula"

Reviewer #2

(Remarks to the Author)

The authors have thoroughly addressed the concerns raised in the initial review, and the revised manuscript is clear and scientifically sound. I recommend it for publication.

Reviewer #3

(Remarks to the Author)

Most of my concerns have been addressed, however there are still two that remain. I have listed them with the original comments, the author responses, and then my continued concern.

18. Line 167-169: There is quite a broad range of IC50's seen. Please discuss. Does this correlate with the affinity of ACE2 for the variant RBD's? Suggest adding a correlation analysis to the data set and add to manuscript figure or supplement.

Author Response:

As pointed out by the reviewer, we measured IC50s between 1937 nM (Delta) and 95 nM (Omicron BA.2) for B07 and between 1180 nM (Delta) and 220 nM (BA.2) for B09. Several reports showed that the affinity of RBD for hACE2 tends to

increase with new variants (for example, Pham Dan Lan, 2024; PMID: 38310477). The measured IC50s of B07 for the different variants could be correlated with the affinity of hACE2 for RBDs. However, we did not find data comparing hACE2: RBDs' affinities for the different variants tested in our study. We consider this point too speculative to be mentioned.

Reviewer Response:

Because the VHH is kinetically competing for binding with the RBD for ACE2, a variant correlation analysis would improve the quality of the paper. This is because we would expect that a variant RBD with better affinity for ACE2 than the VHH would result in a worse variant-specific VHH IC50 and vice versa. Readers of this paper will be familiar with the concept that ACE2 receptor affinity is changing with variants, as it has been discussed and evaluated many times in the literature. In addition to what was mentioned, a quick search came up with PMID:35093192, 35324257, 36399443. Thus, as a reader, I would not think this is a speculative question to be asking. It would be a question that I would expect to see analyzed.

25. Figure 5 legend: MOI and hours of the assay readout are not the same between viruses for replication-based neutralization assays. This can cause the observed differences in IC50 values being reported between variants and B07-Fc. A higher MOI means more virus is present and therefore more spike proteins. The consequence of this is that more spike proteins are present on particles that need to be neutralized and the result is a shift to a worse IC50 relative to a lower MOI virus. Similarly, increase assay readout time means that additional replication cycles are occurring and this results in the generation of more virus genome copies. Since the assay is measuring viruses by RT-qPCR, it would mean that there could be a falsely worse IC50 value for B07-Fc for viruses assayed at 66 hours vs those assayed at 48 hours. The authors need to provide adequate justification of varying the assay MOI and time differences and provide evidence that the differences do not affect IC50 values for each variant. In the absence of that, they will need to report the comparisons done under the same conditions to avoid biases.

Author Response:

MOI and incubation time were defined based on previous studies in order to obtain a similar rate of viral production after infection with the different isolates. We clarified this point in the Methods section (Line 758) (Touret, 2023 ; PMID: 36968074).

Reviewer Response:

Neutralization is a measure of the impact of an intervention on the virus entry. It is therefore highly impacted by changes in the number of virus particles present. This is because particle number is directly correlated with the number of spikes that are available to interact with ACE2. More spikes provide a higher probability that ACE2 will become bound and entry will occur when ACE2 is not being occupied by the VHH.

Using the rate of viral production after infection is not appropriate because it is a multifactorial output that is primarily reflective of the rate of viral replication and budding that occur after entry. For example, a poorly entering virus may compensate by having a higher rate of replication. Therefore, the justification to vary the multiplicity of infection (MOI) to account for replication differences is not sufficient because it can lead to inaccurate conclusions due to confounding post-entry events.

If replication time is not a concern, then a reporter gene-based, immunofluorescence, or RT-PCR approach are all acceptable as long as the MOI's are the same. However, based on the response, it seems that replication times are a concern.

If the MOI is kept constant and there remains a concern for differences in replication time, there are several approaches that can be taken. The first is to perform a plaque assay. With a plaque assay, slower replicating virus will have the same number of plaques as a faster one, they would just differ in the size of the plaque. A less technically challenging approach is to use single-cycle pseudotype viruses. These are well accepted in the field and used commonly to assay antibody neutralization. Lentiviral systems are most commonly used but there are other retroviral and VSV-based single-cycle systems.

Either way, modifying the MOI and incubation time to account for replication differences is not an appropriate way to perform a neutralization assay. Therefore, the neutralization assays need to be redone as either pseudotype entry assays or plaque assay.

Dr. Anne BRELOT
Unité Dynamique des Interactions hôte-Pathogène
Institut Pasteur
25 rue du Dr. Roux,
75724 Paris Cedex 15, France
Tel: +33 (1) 40 61 34 67
E-mail: anne.brelot@pasteur.fr

Answers to the reviewers

We would like to thank the three reviewers for their detailed review of our work. Please find our point-by-point responses below:

REVIEWER COMMENTS

Reviewer #1 (Remarks to the Author):

Summary:

The manuscript developed a pan-SARS-CoV-2 neutralizing antibody B07-Fc for intranasal administration. The paper presents a methodical and well characterization of the VHH starting from its production, panning selection, and detailed in vitro binding assays, including binding affinity studies and functional competition. The authors also effectively demonstrate that these VHH do not interfere with ACE2 enzymatic activity, which is crucial for its development as a therapeutic candidate that will not influence its biological role. The authors have tested B07 across a range of SARS-CoV-2 variants, highlighting the potential of B07 to be effective against a broad spectrum of viral mutations. The author shows that the fusion of VHH candidate B07 to an Fc domain works effectively, enhancing the therapeutic potential by improving binding affinity and functional efficacy. Finally, the authors evaluated the neutralization efficacy of B07-Fc in the K18-hACE2 mouse model and showed that B07-Fc protect against SAR-CoV-2 transmission to lung in three days post viral challenge. However, a long-term mouse study including multiple additional study are required to conclude the antibody efficacy and stability. Since intranasal administration is most effective against initial viral infection, the author should emphasize more on the significance of intranasal administration compared to conventional approach. In conclusion, while the manuscript presents promising findings, significant issues weaken the study's overall impact. The lack of robust statistical analyses and insufficient in vivo efficacy data undermine the comprehensiveness of the study.

Major comments:

1. In Fig 2B, the number of data points of B09 and B10 groups (only 2 samples) are different than ctrl and B07 samples (5 samples).

We corrected Fig. 2b by adding data points for groups B09 and B10, which were missing. Now each bar corresponds to the mean +/- SD of at least 3 data points. The overall results remain unchanged.

2. Authors should show statistical analyses (p-values confidence intervals (CIs) for VHH IC50 values, and R² values for dose-response curves) on the figures.

We added the confidence intervals of the p-values for the VHH IC50 and R² values in **Supplementary Table 1**.

3. The flow data in Fig 2C is inconsistent with the quantitative data in Fig 2B. From flow, geo mean of B09 binding to IGROV-1 show a higher shift compared to B07 while the quantitative data show the other way around.

We thank the reviewer for this comment. We have added new flow images based on our new dataset (Fig. 2b, see above). Flow and quantitative data are now consistent regarding the IGROV-1 cell type.

4. Fig 3B, C, Fig 5E lacks mock control to set the 100% enzymatic fluorescence baseline.

The parameters to achieve the 100% enzymatic fluorescence baseline were defined after optimizing the soluble hACE2 concentration and the time required to reach a plateau of fluorescence, corresponding to maximal peptide cleavage. Based on the “*Figure Kinetics*” below, we used sACE2 at 10 nM for 30 min.

Figure Kinetics: Kinetics of cleavage of a fluorogenic peptide substrate depending on sACE2 concentration using SensoLyte 390 ACE2 Activity Assay kit (AnaSpec).

In Fig. 3B, C, Fig. 5E, the mock control (buffer) corresponding to the 100% enzymatic fluorescence baseline was arbitrarily placed as the lowest concentration data point. To avoid any misunderstanding, we have now positioned the mock control point at an “extremely low” concentration (10^{-14}) in the Fig. 3b, c, new Fig. 6e. This is specified in the legend of Fig. 3 (Line 1268 and 1270).

5. In Fig 3D, Author should briefly explain why fluorophore (ACE2 enzymatic activity) increased.

We added one set of data for each condition of the Fig. 3D. A statistical analysis showed a non-significant impact of B07, B09, and B10 on sACE2 enzymatic activity. Statistics are now showed in Fig. 3d. Overall, this experiment showed that all three VHHs did not reduce the enzymatic activity of sACE2, which is crucial for a therapeutic use of the antibodies.

6. Fig 5G, to demonstrate the inhibition of SARS-Cov-2 infection by selected VHHs, an assay that directly shows the reduction of infected cells may be more relevant. For example, use GFP-labeled SARS-CoV-2 to infect the VHHs pre-incubated cells. Adding spike protein immunostaining in the S-Fuse assay is also recommended.

As requested by the reviewer, in addition to the S-Fuse assay, and to further validate the effect of B07-Fc on infection, we directly measured the number of infected IGROV-1 cells in the presence or absence of B07-Fc by anti-nucleocapsid immunostaining. We added a new figure (new Fig. 6h), confirming the inhibitory activity of B07-Fc for two recent variants (JN.1.1, KP.3.3) (IC₅₀ of 0.004550 μ g/mL and 0.005981 μ g/mL, respectively, corresponding to 0.06 nM and 0.07 nM) (Line 311). For consistency, we also measured by S-Fuse the effect of B07-Fc on the inhibition of these two variants (new Fig. 6g, Line 307). The lower IC₅₀ measured in the infection assay compared to S-Fuse is explained by the low amount of hACE2 expressed at the surface of IGROV-1 cells compared to U2OS cells used in the S-Fuse assay. We detailed the infection assay in the Methods section (Line 730).

7. In Fig 6B, since mouse and hamster ACE2 share identical sequence, why B07 bind to hamster and mouse ACE2 different affinity?

We thank the reviewer for this remark. We made a mistake in typing the hamster ACE2 sequence. We have now corrected it in Supplementary Fig. 9b. The hamster and mouse ACE2 sequences between residues 27 and 42 are different at position 30-31. Residues NN30-31 in the mouse sequence likely explains the lower affinity of B07-Fc for mouse ACE2 compared to hamster ACE2. We explained this on Line 343. To further address the impact of this genetic variation on B07-Fc binding, we introduced NN residues at position 30-31 in human ACE2. The corresponding mutant (DK30-31NN) is less recognized by B07-Fc compared to the WT hACE2, confirming this assumption (Figure DK-NN below).

Figure DK-NN: B07-Fc binding on cells expressing myc-hACE2 WT or myc-hACE2 DK30-31NN. Transfected cells were incubated with an anti-myc antibody (Abnova, 1/500) or B07-Fc (0.1 $\mu\text{g/ml}$) and stained with a AF647-conjugated anti-mouse antibody or a AF647-conjugated anti-human antibody respectively, before being analyzed by flow cytometry. Data are means \pm SD of three independent experiments.

8. In Fig 6B, structure or docking prediction of B07 VHH interaction with human ACE2, hamster ACE2, mouse ACE2 is highly recommended.

We determined the X-ray structure of the complex between soluble hACE2, B07, and B10 at 2.7 \AA resolution. The overall structure reveals that B07 binds an epitope mainly involving the hACE2 helix α 1, with all three CDRs of B07 providing contacts. We have added new Figures (new Fig. 5, Supplementary Fig. 4-7, Supplementary Table 2-5) presenting this structure. Note that following the addition of a new figure (new Fig. 5), the old Fig. 5 now becomes Fig. 6. We mentioned these new results (Lines 243-283) and the methods (Lines 780-821). We validated B07 epitope on hACE2 by exchanging residues at positions 34, 35, 38 of hACE2 by an alanine. The respective mutants abrogated B07 binding on hACE2 expressing cells. Comparison of the 3D model of B07:hACE2 with the RBD:hACE2 models showed clashes between B07 and RBDs (SARS-CoV-2, SARS-CoV-1, NL63), explaining the competitive inhibition mechanism of B07 (Supplementary Fig. 6). B07 will also block potential emerging human-adapted SARS-CoV-2 derived from mink (Supplementary Fig. 6 and discussion on Line 428).

9. The animal study in Fig 7 is not sufficient to support the conclusion that “Fc-conjugated B07 dimers protect against Omicron infection in vivo”. In Fig 7A, the study only observed infected mice for up to 3 days. The study should include a long-term evaluation (up to 14-21 days). This also help to assess B07-Fc half-life and strengthen the claim of B07-Fc protection against viral infection.

We agree that a long-term study would allow assessing VHH pharmacokinetic parameters, such as its half-live, and that evaluation of mice protection after administration of B07-Fc against lethal infection requires a 2 to 3 weeks follow-up. However, infection of K18-hACE2 mice with Omicron

XBB.1.5 causes attenuated or no disease in mice (PMID: 35062015). Therefore, we performed a new experiment where we infected K18-hACE2 mice with a lethal SARS-CoV-2 variant (D614G isolate) (Fig. 7f-h). This model is recognized as very severe.

Following intranasal administration of this variant, control mice (ctl-Fc) reached humane endpoint (body weight loss >20% or clinical score >6) on day 6 post-infection (pi) and were euthanized (Fig. 7f). Data in the B07-Fc group were heterogeneous, with some mice minimally affected (weight loss < 10% at day 6 pi) while others exhibited the same severity as ctl-Fc mice. Based on the results of the first experiment, we euthanized all B07-Fc mice to measure VHH concentration in the lung. As before, we observed that mice with a high concentration (> 200 ng/g of lung) were mildly affected (body weight loss < 10%), while the body weight loss curve in mice with lower concentration was similar to that of ctl-Fc mice (Fig. 7g, h). Although the follow-up of infected mice was interrupted to evaluate VHH lung concentration, these results suggest that full protection could be obtained if the delivery method of VHH was improved to achieve consistently high lung concentrations. This is explained on Lines 372-388.

10. In Fig 7B, as previously mentioned, the manuscript does not provide statistical comparisons between the control (ctl-Fc) and treated (B07-Fc) groups. Only 6 out of 10 mice showed a reduction in viral load is not convincing to demonstrate the B07-Fc protection. Author explanation of the variability is due to heterogeneity. However, if the variability is due to the sub-optimal distribution of B07-Fc in the lungs, the authors should propose or implement methods to improve this distribution, ensuring consistency across the treated animals. If the variability is potentially due to VHH degradation or aggregation, it is important to conduct tests to assess these factors.

The heterogeneity of the B07-Fc group resulting from inconsistent biodistribution of VHH in the lung prevented statistical analysis from using this group as a whole. Rather, the sigmoid shape of the distribution in Fig. 7e strongly suggests a dose-effect response.

We agree with the reviewer that the efficient use of our VHH would require improving the delivery method to ensure achieving consistently high lung concentrations. We used intranasal administration which does not require specific instruments. A 10 µl drop placed at the nostrils opening must be inhaled by the sleeping mouse. However, the complex anatomy of the nasal cavity of mice impacts the diffusion of the inhaled antibodies (PMID: 39711323). Therefore, it may result in heterogeneous distribution of the molecules. We acknowledge that this is an inherent limitation of this model. Achieving homogeneous distribution would imply the administration of B07-Fc after atomization in micrometric range liquid droplets, using aerosolization or nebulization (PMID39711323). We discussed with an expert in the field of monoclonal antibody administration in the respiratory tract in animal models, and he emphasized the complexity of the technique that require specific equipment and several months of training and validation before obtaining satisfactory results. In addition, since we are infecting mice, this requires to be carried out in a BSL-3 environment which adds to the complexity. While these experiments are theoretically feasible, we considered they were beyond the scope of this article.

11. Compared to qPCR-based titration assay shown in Fig 7B, a conventional plaque assay may be more relevant to actual viral titer.

We performed the plaque assay titration in parallel with the qPCR-based assay. Since the result was identical, we only showed one of the two assays. As requested by the reviewer, we have now added a new Fig. 7c, showing the results of the plaque assay. These results confirm the result of the qPCR-based titration assay. We mentioned the new figure on Line 354.

12. Fig 7C, it would be more direct to see lung infections if the author shows immunostaining of the SARS-CoV-2 virion particle.

We have now added immunostaining of the SARS-CoV-2 nucleoprotein (N) to investigate the impact of B07-Fc on infection (new Fig. 7d and Supplementary Fig. 10). N protein was detected in nasal respiratory epithelium, olfactory epithelium and lung bronchioli in VHH-ctl treated cells while it was undetectable in B07-Fc-treated cells 3 days pi. This suggested that VHH B07-Fc protects cells from infection, in addition to protect them from tissue damages. We mentioned this new result on Line 358 and completed the Methods section (Line 840).

13. Fig 7. Infection of K18-hACE2 mice with SARS-CoV-2 can lead to lethality. Therefore, to better illustrate intranasal B07-Fc protection from lethal challenge, a mouse survival curve should be included. It would be also interesting to see the immune profile of the challenged mice.

As mentioned above, infection of K18-hACE2 mice with Omicron SARS-CoV-2 variants do not lead to lethality. Lethal models can be induced using earlier variants, such as Wuhan, D614G or Delta variants. They have been used, including by us, to test vaccines and monoclonal antibodies (C. Planchais, 2022, PMID:35704748; P. Brandys, 2022, PMID:35874687; C. Planchais, 2024, PMID:39071888). They induce death in 6-8 days and are considered overly severe compared with human pathology.

As presented above in point 9, we used a D614G variant to infect K18-hACE2 mice and could show that B07-Fc was able to reduce the severity of infection to a point where mice would have very likely survived after 2-3 weeks, based on our experience of hundreds of SARS-CoV-2-infected mice. However, we euthanized the mice earlier to measure the VHH concentration in their lung in order to establish a dose-effect curve as presented in Fig. 7h.

Measuring the immune profile in B07-Fc-treated and ctl-Fc mice would provide additional parameters to assess protection (suppression of lung inflammation) but not on the mode of action of B07-Fc which specifically targets ACE2.

14. Fig 7. Since the neutralizing antibodies were produced using the E.coli system, it seems necessary to supplement experiments regarding the removal of endotoxins such as LPS. For example, in the case of in vitro modeling, the study was conducted based on HEK293 cells, which have minimal expression of toll-like receptors, so there might be insufficient validation of potential side effects. Furthermore, in the animal model, there are only results for intranasal administration. Because intranasal only protect viral early intranasal infection, and stability of B07-Fc was not assessed, it seems necessary to add cross-validation for other commonly used administration methods such as IP or IV injection, as well as data on the degradation of neutralizing antibodies within the animal body.

VHH B07-Fc, which is used *in vivo*, was produced in Expi293F cells and not in an E. Coli system (unlike monomeric VHHs, which were produced in E. coli TG1 strains). This point is mentioned in the Methods section (Line 647).

As proposed by the reviewer, future developments include assessing the pharmacokinetics of VHH after IP or IV administration. Unfortunately, due to the current shortage of K18-hACE2 mice in our breeding facility, we were unable to perform such experiments in the time allowed for revising the manuscript.

Minor comments:

1. In introduction section, pan-neutralizing antibody can be also achieved by directly targeting the RBD of the S protein from SARS-CoV-2 other than anti-hACE2 antibody that the author proposed. Both strategies have its own advantages and disadvantages. The manuscript should include this explanation and highlight the study novelty.

Targeting S has the advantage to directly neutralize the virus entry. However, the emerging Omicron subvariants have a remarkable ability to spread and escape nearly all current monoclonal antibody (mAb) treatments targeting the S protein RBD (Planas, 2024, PMID: 39391808). Two strategies have been proposed to suppress antigenic escape and achieve a pan-neutralizing antibody: mAb cocktails and bispecific antibodies. While both can minimize escape, neither is currently used in clinical use due to manufacturing constraint to produce these antibodies in time with the course of infection. We mentioned this in the Introduction section (Line 94).

In comparison, targeting a host protein presents a lower risk of viral escape, but a higher risk of altering the activity of the targeted protein. Given the role of hACE2 in blood pressure regulation, identifying this risk is paramount. If this risk is eliminated, targeting hACE2 would be advantageous since anti-ACE2s act as pan-antibodies. We showed here that anti-ACE2 VHHs have no impact on ACE2 enzymatic activity *in vitro*, which is promising (Fig. 3, Fig. 6e, f).

2. In Supplementary Figure 2, the incubation time with B07-Fc is not specified.

We have now specified the incubation time (O/N at 4°C) in the Methods section (Line 827).

3. Supplement Fig 3B, the fluorescence voltage should be adjusted to place the control population more towards center.

We now presented these results in bars instead of graphs (Supplementary Fig. 9c). However, as requested, we paid attention to place the control population towards the center (see Figure Overlays below).

Figure Overlays: HEK293 cells transfected with plasmids encoding myc-hACE2-WT or the indicated mutant were stained with an antibody against the myc epitope or B07-Fc (0.1 µg/ml) and analyzed by flow cytometry. Background (Blue) corresponds to the fluorescence intensity obtained using the secondary antibody alone.

4. Intranasal administration dosages in hamsters (Supplement Fig 4, 5mg/kg) and mice (Fig 7, 7mg/kg) are different, which may count for the VHH neutralization efficacy difference.

Although we agree that the difference in VHH dosages used in hamster (5 mg/kg) and K18-hACE2 mice (7 mg/kg) may count for the difference in VHH inhibition efficacy between the two species, we

believe that this difference is mainly due to the presence of a glutamine at position 34 in hamster ACE2 (instead of a histidine in hACE2) for the following reasons:

- The resolved X-ray structure of the hACE2/B07/B10 complex revealed that the residue of hACE2 at position 34 is part of B07 epitope (new Fig. 5).
- B07-Fc did not react with hACE2 H34Q, while introduction of a histidine at position 34 in hamster ACE2 (hamster ACE2 Q34H) restored B07-Fc binding to hamster ACE2 (Supplementary Fig. 9c).
- BLI experiments showed lower affinity of B07-Fc for hamster ACE2 compared to human ACE2 (Fig. 6a and 6k)
- K18-hACE2 mice treated with 5 mg/kg B07-Fc and infected with D614G were protected from body weight loss at day 6 post-infection if they received a sufficient VHH concentration in the lung (Fig. 7g, h).

5. *Authors should explain the rationale of using clone B07 instead of B09.*

We used clone B07 instead of clone B09 because it is more stable at 4°C and more efficient to detect cell surface hACE2 (Fig. 2, new Supplementary Fig. 3) and to inhibit Omicron variants (Fig. 4a) than clone B09.

6. *Standardize font use. The font used in the table in Fig 4A, legend in Fig5 A, table in Fig 5F and G etc. all use different font than others.*

We have now standardized all font.

7. *Fig 7C fluorescence have similar color, which make the graph hard to read. Use of distinct wavelength fluorophores is highly recommended.*

We have now added a Supplemental Figure (new Supplementary Fig. 10) presenting each staining separately and in black and white to clearly distinguish each fluorophore.

8. *Authors need to refine figure titles to better conclude the overall context of figures.*

As proposed by the reviewer, we have now refined some of the figure titles (Fig. 1, Fig. 2, Fig. 4) to better conclude the overall context of figures.

Reviewer #2 (Remarks to the Author):

In this manuscript, Blachier et al. present a study on a novel approach to prevent SARS-CoV-2 infection using single-domain antibodies (VHHs) that target the ACE2 receptor, rather than the spike protein. This strategy is particularly relevant given that some SARS-CoV-2 variants have developed resistance to vaccines and therapies targeting the spike protein. The authors demonstrate that targeting ACE2, the critical entry receptor for the virus, can prevent infection of multiple SARS-CoV-2 variants, including recent Omicron subvariants.

The authors identified and characterized three VHHs (B07, B09, and B10) isolated from an alpaca immunized with ACE2. These VHHs bind ACE2 with high affinity. Two VHHs (B07 and B09) prevent entry of multiple SARS-CoV-2 variants through a competitive mechanism that prevent virus attachment to host cells. In contrast, VHH B10, does not inhibit viral infection, underscoring

the importance of epitope specificity in preventing virus attachment. None of the characterized VHHs affect the catalytic activity of ACE2, which is essential for maintaining normal physiological function.

The authors create dimers by fusing the VHHs to an Fc domain, resulting in improved binding and prevention of viral entry.

Overall, the study is well written and well conducted and highlights an interesting concept to prevent infections.

I am surprised the authors call the mechanism neutralization even though the VHHs do not target the virus, but the host. The VHHs prevent entry and infection, but they do not per se neutralize. I recommend adjusting the wording to describe the function of the VHHs more accurately.

Otherwise, I recommend the manuscript for publication.

We thank the reviewer for this constructive remark. We agree that the term neutralization is confusing since it is frequently used as virus neutralization instead of virus entry/replication neutralization. We now refer to “entry inhibition” or “fusion inhibition” or “replication inhibition” to mention the VHH action. The title of the manuscript was changed consequently.

Reviewer #3 (Remarks to the Author):

This manuscript reports the work of Blachier, et al to identify a single-domain antibody (VHH) derived from an alpaca as a potential therapeutic antibody against SARS-CoV-2 (SARS2). ACE2 is an essential protein for the entry of SARS2 and is considered to be the receptor that triggers the viral spike protein (S) to undergo conformational changes that result in the fusion of the viral membrane with the target cell membrane. This is the penultimate step in virus entry (uncoating being the last) and is essential for SARS2 virus infection. Thus, inhibition of ACE2 binding by S is a prominent target of host immune responses, SARS2 vaccines and SARS2 therapeutic antibodies. Because of host immune responses to previous infection and vaccination, SARS2 has undergone significant evolutionary changes in its spike protein, leading to escape from both existing immunity and therapeutic antibodies targeting S. To potentially mitigate against this, the authors chose to target the host cellular protein ACE2 because the number of allelic differences are small and the much smaller rate of evolutionary change in human ACE2 make it a stable therapeutic target.

Single-domain antibodies or nanobody are purported to have advantages over “typical” IgG antibodies (mAbs) that have both heavy and light chains, with most of the theoretical benefits relating to size and their ability to have long CDR3 regions. However, there is a relative lack of data on their clinical use and benefits over classical mAbs and their small size means they are likely to have pharmacokinetic properties similar to that of antigen binding fragments (Fabs) that are derived from mAbs (i.e. have very short half-lives). In addition, they often lack the benefit of the avidity provided by the multivalency of traditional IgG mAbs and the ability to induce antibody mediated effector functions that require an Fc domain. To overcome these issues, VHH’s are often fused to Fc domains to create bivalency, add effector functions and increase their half-lives.

In this report, the authors identify 3 VHH (B07, B09, B10) that target the ACE2 binding site that is bound by S and perform a limited evaluation of their epitopes and ability to block the fusion step in virus entry for multiple variants of SARS2. They perform a more in-depth analysis of their leading candidate B07 and fuse it to an Fc domain to provide enhanced avidity and make it more IgG like. They subsequently use a humanized ACE2 mouse model to show that there is likely a benefit when given as a prophylactic and intranasal therapy. Data in hamsters was not as robust, but they show that this may be due to a significant difference in the ability of B07 to bind to hamster ACE2.

Overall, the approaches are sound, the experiments are done well and the results generally support the conclusions and claims that are made. There are a few issues that need to be addressed and some

conclusions that need to be tempered.

Concerns to be addressed:

1) Line 88. VHH are not fragments as they are complete naturally smaller proteins.

A VHH corresponds to the variable region of camelid heavy-chain antibody. We replaced the term “fragment” by “VHHs” as request (Line 119)

2) Line 90. While it is true that VHH can be produced in prokaryotic systems, the use of an Fc domain abrogates this benefit.

We deleted the sentence to be in line with our main message concerning B07-Fc activity (Line 120).

3) Line 93. “Therefore, they can be delivered to the lung by inhalation, which can confer significant advantage for SARS-CoV-2 treatment.”. It is not clear how the properties described in the preceding sentences means that VHH have properties that allow it to be delivered to the lungs by inhalation where other therapies cannot -- especially if the VHH has been fused to an Fc. Traditional antibodies have also been delivered to the lungs by inhalation and the intranasal route.

We agree with this remark. We rephrased the sentence: « These features enable the development of inhalable preventive formulations, ensuring direct delivery to the lungs, thereby conferring significant advantages for the treatment of respiratory diseases” (Line 124).

4) Line 111. “constructed a cDNA library containing about 8×10^7 different VHHs sequences. The method only discusses PBMCs but not how many B cells went into the library and the method estimates a library size of 8×10^7 cfu (colony forming units?) and the text says VHHs sequences. Is it cfu or sequences? If the later, how was that number estimated. Was the whole library sequenced? And if sequenced, how many unique lineages were represented in 8×10^7 sequences. If it is cfu, then the text needs to be corrected because likely that there are many copies of the same VHH cfu library. Either way, how many input cells were used to create that number of VHHs.

The reported library size of $\sim 8 \times 10^7$ refers to colony-forming units (cfu), not the number of unique VHH sequences. We revised the manuscript to ensure this distinction is clear (Methods section, Line 612-633). Since phage display libraries contain redundant sequences, the actual number of unique VHHs is lower than the total cfu count.

To estimate sequence diversity, we randomly sequenced 50 individual cfu from the library, and all 50 contained unique VHH sequences. While this suggests a high level of diversity, we acknowledge that a larger sequencing effort would be required to determine the exact number of unique VHHs in the full library. The observed uniqueness in our small sample indicates that a significant proportion of the 8×10^7 cfu likely represents distinct VHHs.

The library was constructed from approximately 2×10^8 PBMCs isolated from the immunized alpaca. As VHHs are derived from heavy-chain-only B cells, the actual number of B cells contributing to the library is expected to be a fraction of the total PBMC count. Total RNA was extracted from these cells, and 20 μ g of RNA was used to generate the cDNA library.

5) Line 113-115. “192 individual clones were assayed by ELISA”. If it is known if these were unique sequences in the clones, please provide the number of unique as it will help put the small number of positive clones in context.

We tested 192 individual clones in the phage ELISA, of which 123 were positive. All 123 positive clones were sequenced, revealing that 80 belonged to a dominant cluster derived from the same IGHV3S53 gene as B07 and B09. Within this cluster: B07 was found 31 times; B09 was found 23 times. Given that B07 and B09 originate from the same IGHV3S53 gene and differ by only six amino acid mutations in their sequences, they likely represent variants of the same lineage. B10, in contrast, was found only once and originates from a different IGHV3-3 gene, indicating a distinct evolutionary origin. We detailed this in the Methods section (Lines 612-633) and in the manuscript on Lines 150-157.

6) Line 116-117. If B07 and B10 differ only by 6 CDR amino acids are they from the same lineage? If so, please indicate. Similarly, please provide information on which CDRs and if there are framework mutations.

B07 and B09 share the same IGHV gene (IGHV3S53) and differ by only six amino acid mutations within their sequence. Given this high sequence similarity, they likely originate from the same lineage, arising from somatic hypermutation of a common ancestral VHH. We explicitly stated this in the Methods section (Line 627-633).

Sequence alignment shows 2 mutations between B07 and B09 within CDR1 and CDR2, while CDR3 remains identical. B10, in contrast, is derived from a different IGHV gene (IGHV3-3) and exhibits more substantial sequence divergence in all three CDRs, as well as in some framework regions. We added Supplementary Fig. 1 aligning B07, B09, and B10 sequences, showing their differences.

7) Please provide supplementary figure showing the sequences for each VHH and include an alignment of B07 and B09 showing their differences

We added a supplementary figure (Supplementary Fig. 1) showing the full sequences of the three VHHs highlighting their differences.

8) Lines 119-133/Figure 1. BLI measurements of VHH affinities to soluble ACE2 (sACE2).
a. Please comment if sACE2 was monomeric or dimeric form. It appears to be dimeric and if so, the affinities are likely apparent KD because the VHH is bound to the sensor chip. The model used for curve fitting does not match the raw curves well and this could indicate that sACE2 is a dimer and/or that a 1:1 model may not be the best to use. Please justify the use of a 1:1 fitting model. Alternatively, it would be best to put the sACE2 on the sensor tip and use the VHH as the analyte in solution.

We thank the reviewer for this constructive comment. The soluble ACE2 (sACE2) used in our experiments corresponds to a recombinant construct encompassing residues 19-615, which lacks the C-terminal domain involved in dimerization. Our new structural data support that this truncated form exists as a monomer, both in solution and in the crystal structure. We applied a 1:1 Langmuir binding model to estimate the apparent KD values. While we acknowledge that the raw sensorgrams show some deviation from this model, it provided a reasonable approximation of binding affinities (Karlsson & Fält, 1997, PMID: 9005951; Rich & Myszka, 2008, PMID: 18951413) (Supplementary Fig. 2a).

To address the reviewer's concern more rigorously, we performed new BLI experiments in which ACE2-Fc was immobilized on the sensor tip, and VHHs were injected as analytes in solution. This reversed orientation resulted in high-quality sensorgrams with well-behaved association and dissociation phases. The data were again fitted using a 1:1 model, which provided good fits. Importantly, applying a biphasic (2:1) model did not significantly alter the KD estimate for the

dominant binding phase (contributing ~82% of the signal), supporting the appropriateness of the 1:1 approximation in this context.

We have replaced the original Fig. 1a with these new results and moved the initial data to the Supplementary Information (now Supplementary Fig. 2a), as suggested.

b. Please convert concentrations from ug/ml to nM for the concentrations of sACE2 used.

The concentrations of sACE2 were converted to nM and indicated on Fig. 1a and Supplementary Fig. 2a.

c. Please repeat BLI measurements for B07 and B08 to include a concentration range of sACE2 where the upper value is at least 10-fold more than the measured IC50. This is the standard recommendation to ensure an accurate measurement.

As proposed, we repeated BLI experiments for B07 and B09 in a range of sACE2 concentration where the upper value is 10 -fold more than the previous measured IC50 (new Supplementary Fig. 2). We also repeated the BLI measurement for B10.

9) Lines 124-125. The affinity for B07 and B08 are poor (~350 nM). High affinity is generally single digit nM affinity or lower.

We replaced good by “moderate” affinity (Line 166).

10) Line 135 and later. It would be helpful to the reader to indicate human ACE2 where appropriate because mouse, hamster and other species ACE2 are used in different locations. For example, endogenous ACE2 on Vero cells (Line 138) is a non-human primate.

We now indicated in the text the species of the ACE2 studied: human (h), mouse (m), hamster (ha), and non-human ACE2.

11) Line 140. Does the control VHH also have a myc-tag? If not, assay will need to be repeated using a VHH control with a myc-tag so it matches the experimental VHHs and their detection. If so, please modify text to state that.

The control VHH as well as the anti-ACE2 VHHs are myc- (and His)-tag. It is now mentioned in the text (Line 188).

12) Figure 2 ACE2 surface binding. Please provide a positive ACE2 binding control. These are commercially available to purchase. In addition, please provide a control that tests binding of SARS2 spike or RBD. These will help show the expected binding ranges based on the ACE2 expression levels for the VHHs.

As proposed by the reviewer, we added 2 additional figures evaluating the ACE2 expression level on the different cell lines:

- Supplementary Fig 3b: anti-ACE2 commercial antibody (R&D system, AF933) staining of the different cell lines.
- Supplementary Fig 3c: soluble Spike binding on the different cell lines.

These results are mentioned in the text (Line 185).

13) Line 150. In order to determine if 10 uM is an appropriate amount of VHH to use for an off-

target enzymatic inhibition assay, the manuscript should provide a rationale for choosing 10 μM . The value should be chosen based on an expected C_{max} and steady state concentration of the VHH. They could choose the value based on human or animal studies of VHH therapeutics. Offhand 10 μM seems too low.

10 μM of VHH (165 $\mu\text{g}/\text{mL}$) is 100 times higher than the dose of caplacizumab (10 mg), a bivalent VHH targeting von Willebrand factor, used in human (10 mg might correspond to 2 $\mu\text{g}/\text{mL}$ if 100% of the compounds reaches the bloodstream) (Scully, 2019, PMID: 30625070). 10 μM is also more than 10 times higher than the IC_{50} value ($<1\mu\text{M}$) measured by S-Fuse (Fig. 4). To verify the safety of VHHs with higher VHHs concentration, we repeated the experiment with undiluted VHHs (Figure Activity below). With a concentration higher than 20 μM , we observed no effect of the three VHHs on the enzymatic activity of hACE2.

Figure Activity: Enzymatic activity of soluble ACE2 in the presence of VHHs B07 (66.7 μM), B09 (24.7 μM), B10 (23.3 μM) or absence assayed using the Sensolyte 390 ACE2 activity Assay Kit, which measure fluorogenic peptide cleavage.

14) Inhibition assay. The manuscript would be improved by providing the reader with some basic information about the in vitro and in cellulo ACE2 enzymatic cleavage assays, substrate and positive control inhibitors in the text. Some of the information is present but it needs to be rephrased in a way that brings it all together and adds one or two small missing bits.

We added information to complete the description of the Sensolyte 390 ACE2 enzymatic assay in the Results (Line 200) and in the Methods (Line 707) sections.

15) “Neutralization assays” Lines 160 and below. In neutralization assay, virus particles are incubated with increasing amounts of antibodies or VHH prior to being added to cells. Following a set period, the number of cells that are infected is measured. The differences in number of cells infected when antibody is present, and no antibody is defined as the neutralization. SARS2 entry is a multistep process that includes cellular attachment, receptor binding, fusion and uncoating. In addition, there is also an endosomal entry pathway involving proteolytic cleavage and a cell surface pathway utilizing TMPRSS2 (Nature Reviews Molecular Cell Biology volume 23, pages 3–20 (2022)). Thus, each of these steps can be targeted by an antibody and can influence an antibody activity. Thus, fusion assays are a proxy for neutralization activity and while they can be used to screen for neutralization, the term neutralize or neutralization should not be used as they only model one aspect of entry. Please modify the manuscript and figures labels to reflect the assays as a fusion inhibition assay.

We agree with this comment. We modified the term “neutralization” by “fusion inhibition” in the manuscript and modified the title consequently.

16) While it is reasonable to screen for activity of multiple variants using the fusion assay, it is possible that the VHHs might fall off of ACE2 during transport into endosomes or at the low pH conditions found in endosomes. The authors should first perform a traditional virus or pseudovirus neutralization assay using at least one variant to demonstrate that the neutralization IC₅₀ for the VHHs are similar to that seen in the fusion assay. This would provide the justification to use a fusion assay as a proxy for neutralization of other variants.

We showed that the anti-ACE2 VHHs act through a competitive mechanism, preventing virus-cell interaction at the cell surface, prior to membrane fusion or endocytosis (Fig. 4b-d; Fig. 5). However, as proposed by the reviewer, to ensure that this assay is reliable to evaluate the inhibitory activity of anti-ACE2 VHHs, we directly measured the number of infected IGROV-1 cells in the presence or absence of B07-Fc by detection of the viral N protein (anti-nucleocapsid immunostaining). Two recent variants were tested (JN1.1, KP.3.3) (Fig. 6g). The IC₅₀s measured in this infection inhibition assay confirmed the inhibitory activity of B07-Fc, while being 2 log lower than those measured in S-Fuse. This is likely due to the lower amount of hACE2 present at the cell surface of IGROV-1 cells compared to S-Fuse U2OS-hACE2 cells. Therefore, the potency of VHHs may be underestimated by S-Fuse. These results are mentioned on Line 311.

17) Since it is likely that the VHH will see low pH, the manuscript would be improved by determining KD values for B07, B09 and B-7-Fc to sACE2 at a pH that would be found in endosomes (e.g. pH 5.5).

As mentioned above, VHH binding to hACE2 would occur primarily at the cell surface and not in endosome. While it is true that SARS-CoV-2 can enter cells via endocytosis, this process will occur after ACE2-S binding at the cell surface (Hoffmann, 2020, PMID: 32142651). Therefore, we assume that the VHH will not see low pH. In addition, conducting these measurements using our current Octet biosensor setup presents technical challenges due to the impact of low pH on ligand immobilization. For these reasons, we did not perform such experiments.

18) Line 167-169: There is quite a broad range of IC₅₀'s seen. Please discuss. Does this correlate with the affinity of ACE2 for the variant RBD's? Suggest adding a correlation analysis to the data set and add to manuscript figure or supplement.

As pointed out by the reviewer, we measured IC₅₀s between 1937 nM (Delta) and 95 nM (Omicron BA.2) for B07 and between 1180 nM (Delta) and 220 nM (BA.2) for B09. Several reports showed that the affinity of RBD for hACE2 tends to increase with new variant (for example, Pham Dan Lan, 2024 ; PMID: 38310477). The measured IC₅₀s of B07 for the different variants could be correlated with the affinity of hACE2 for RBDs. However, we did not find data comparing hACE2:RBDs affinities for the different variants tested in our study. We consider this point too speculative to be mentioned.

19) Line 190. “to stabilize the VHHs and increase their efficacy.”

a. What is meant by stability and how is being in an IgG form going to help with stability? Up until this point stability of the VHHs has not been mentioned. Please speak to what the stability issue is and how IgG1 fusion solves this. Alternatively, remove the word stability.

We thank the reviewer for this comment. Indeed, Fc fusion increases avidity (Schepens et al., 2022, PMID: 34609205) and half-life in circulation (Ulrichs et al., 2011; PMID: 21576702), not VHH stability. We removed the word “stabilize” (Line 287).

b. Since animal efficacy of the VHH vs the Fc-VHH forms is not studied, efficacy improvement is evaluated in this report. I would recommend changing to potency because that is being measured and is shown to be improved. Alternatively, efficacy can be removed.

We agree with this comment. We removed the word efficacy (Line 288).

20) Lines 205-209. T

a. For the comparisons of IC50s in the fusion assay it is confusing to have a dash between the conversion of ug/ml to molar because it seems that the ug/ml is the VHH range and the nM is the VHH-Fc range. I would recommend removing the ug/ml range and adding in the nM converted B07 BHH result in its place.

b. Similarly, I would show the number for B07-Fc and for the VHH version of B07 as nM.

We now mentioned the IC50s for B07 and B09 in nM (Supplementary Table 1b). As suggested, we now compared the IC50 range for B07 and B07-Fc (in nM) (Line 308).

21) Binding efficacy Section (Lines 214-233). The investigations of allelic polymorphism with the vicinity of the S binding site of ACE2 is a good experiment. But the best experiment would be to obtain a more accurate model of the location of B07 binding to determine which residues to test. A negative stain electron microscopy 3D reconstruction (NSEM-3D), deuterium exchange or full structure, would be able to provide a better footprint and provide mechanistic insights. NSEM-3D is relatively quick to get and when coupled with fitting models of ACE2 and a VHH could provide insights into the mechanism of action and candidate binding residues to test in ACE2. For example, comparing the 3D model with Spike:ACE2 models could inform whether competition is direct or indirect (i.e., does B07 block ACE2 by binding directly to S binding sites or is it a steric blockade). I think it is reasonable to obtain NSEM-3D.

As mentioned above in response to Reviewer #1 point 8, “we determined the X-ray structure of the complex between soluble hACE2, B07, and B10 at 2.7Å resolution. The overall structure reveals that B07 binds an epitope mainly involving the hACE2 helix α 1, with all three CDRs of B07 providing contacts. We have added new Figures (new Fig. 5, Supplementary Fig. 4-7, Supplementary Table 2-5) showing this structure. We mentioned these new results Lines 243-283 and the methods Lines 780-821. We validated B07 epitope on hACE2 by exchanging residues at positions 34, 35, 38 of hACE2 by an alanine. The respective mutants abrogated B07 binding on hACE2 expressing cells. Comparison of the 3D model of B07:hACE2 with the RBD:hACE2 models showed clashes between B07 and RBDs (SARS-CoV-2, SARS-CoV-1, NL63), explaining the competitive inhibition mechanism of B07 (Supplementary Fig. 6). B07 will also block potential emerging human-adapted SARS-CoV-2 derived from mink (Supplementary Fig. 6 and discussion Line 428)”.

22) Line 221. “correctly detected” is subjective. Please change this to something more objective or quantitative.

We removed “correctly” and mentioned that “None of the substitution abrogated the cell surface expression of myc-hACE2 constructs” (Line 325).

23) To better support claims of breadth and potency across time, more contemporary spikes should be evaluated in the fusion assay (assuming it is shown to be equivalent to neut) or neutralization assay.

We added new figures (new Fig 6g, h), confirming the inhibitory activity of B07-Fc for two recent variants (JN.1.1, KP.3.3) tested both by S-Fuse (Fig. 5f) and by measuring the number of infected cells (viral N detection) (Fig. 5g).

24) Discussion:

a. There is a lot of emphasis on VHH advantages when the deliverable here is VHH-Fc which loses most of the advantages relative to a native IgG mAb. Limitations of both approaches should be addressed in the discussion.

Although the VHH-Fc format loses some of the advantage of small size of monomeric VHH, it retains several benefits over conventional IgG mAbs: their smaller size than IgGs (~80 kDa vs. ~150 kDa for IgG) improves tissue penetration; their production from a single-gene, reduces production cost. We clarified these points in the discussion (Lines 443-444).

b. The report is not the first to target ACE2. Please place this research in the context of previously reported work and how this effort is better and/or overcomes limitations of previous reports.

Five previous articles described mAbs targeting hACE2 (Zhang, 2023 PMID: 37188812 ; Chaouat, 2022 PMID: 35992307 ; Du, 2021 PMID: 34404805; Chen, 2021 PMID: 34433803 ; Sun, 2023 PMID: 37689971; Redd, 2024 PMID: 38925442). However, this is the first report of a VHH anti-ACE2 and the first to test the activity of an anti-ACE2 after intranasal administration. We precise these points in the discussion (Line 402; Lines 432-444; Line 460).

c. Line 277. “Resistance would require a change in the way S binds to the ACE2/VHH complex, likely incompatible with the fusogenic activity of the protein”. This is not necessarily the case, as the viral spike could increase its affinity to ACE2 and thus outcompete the ability of the VHH to bind to ACE2. It might do this by making it off-rate much slower than that of the VHH.

We agree. We removed the sentence (Line 420).

d. Line 309. “This would lead to synergistic effects.” To strong, suggest changing would to may.

We tempered our purpose by using the word “potentially” (Line 447).

e. Line 320: “high efficacy” is used when discussing affinity. Change efficacy to affinity.

This is a mistake. We replaced “efficacy” with “affinity” (Line 433).

25) Figure 5 legend: Moi and hours of the assay readout are not the same between viruses for replication-based neutralization assays. This can cause the observed differences in IC50 values being reported between variants and B07-Fc. A higher moi means more virus is present and therefore more spike proteins. The consequence of this is that more spike proteins are present on particles that need to be neutralized and the result is a shift to a worse IC50 relative to a lower moi virus. Similarly, increase assay readout time means that additional replication cycles are occurring and this results in the generation of more virus genome copies. Since the assay is measuring viruses by RT-qPCR, it would mean that there could be a falsely worse IC50 value for B07-Fc for viruses assayed at 66 hours vs those assayed at 48 hours. The authors need to provide adequate justification of varying the assay moi and time differences and provide evidence that the differences do not affect IC50 values for each variant. In the absence of that, they will need to report the comparisons done under the same conditions to avoid biases.

MOI and incubation time were defined based on previous studies in order to obtain a similar rate of viral production after infection with the different isolates. We clarified this point in the Methods section (Line 758) (Touret, 2023 ; PMID: 36968074).

26) Figure 1B.

a. upper left panel.

i. It is confusing to show mock data as a schematic of the result as a panel. I recommend moving the panel to the supplement and replacing it with a cartoon description in the main text or supplement. Also, the title of the schematic panel should clearly indicate it is a schematic by saying “Schematic of” or something similar.

We removed the “schematic with mock data” from Fig. 1b. Instead, we added a cartoon description of the experiment in the supplementary information (Supplementary Fig. 2b). We also clarified the title.

ii. Since ACE2 is bound to the sensor chip, it is confusing to have ACE2 in the title. I would just say VHH competition assay or VHH epitope mapping

We changed the title of the Fig. 1 by “VHHs binding on soluble ACE2 and epitope mapping”.

b. What is being assessed is the ability of one VHH to block the binding of another VHH, so having ACE2 in the title is potentially confusing for a reader. I recommend changing other subpanel titles to “[VHH name] as competitor”. For example, bottom left is “B09 as competitor”.

We apologized for the confusing figures. We now specified in Fig. 1b which VHH is the VHH1 and which one is the VHH2 (competitor).

27) Figure 2 and others. Please provide an example gating tree for all flow experiments in the supplement.

As requested by the reviewer, we provided an example gating tree for flow experiments (Supplementary Fig. 3a, and supplementary Fig. 7a).

28) Figure 4 and elsewhere. Because the assay is a measure of fusion and not neutralization, please change “Neutralization (%)” to “Fusion inhibition (%)” or something similar.

We changed “neutralization” by fusion inhibition in all figures where it was necessary (Fig. 4a; Fig. 6g).

29) Figure 4B/C/D and elsewhere. Since multiple variants are used in the paper, please indicate the variant spike name used in the graph label and/or figure legend text.

The variant spike was specified in the legend of Fig. 4.

30) Figure 5A and elsewhere. Please ensure that BLI graphs have all the individual concentrations indicated (in molar) for each curve in the graph or legend (i.e., a range of concentrations or ug/ml is not acceptable).

We specified the individual concentrations of B07-Fc in nM in new Fig. 6a.

31) Figure 5G.

a. Recommend moving the curve key from the XBB.1.16.1 panel to the far-left panel (i.e. Delta) since most readers read from left to right.

We now presented Fig. 5g in the Supplementary information (Supplementary Fig. 8e). We moved the curve key to the right side of the Figure.

b. If I am correct, “Viral replication inhibition” percent would approach 100% at low concentrations and “(% of untreated cells)” would be 0% at low concentrations. The y-axis is labeled with both. I believe the latter is what is desired based on the data shown. Need to correct this.

We thank the reviewer for the comment. We corrected the y axis of Fig. 5G (now Supplementary Fig. 8e).

32) Figure 6

a. Panel C (BLI).

i. Please provide the exact concentrations in nM for each curve in the panel or legend (as opposed to a range).

We provided the exact concentrations in nM in new Fig. 6k.

ii. In addition, the experiment needs to be repeated to determine if the 65 nM apparent K_D value is correct because:

1. Only 3 sensorgrams are provided and 4 to 5 is the generally accepted minimum to have an accurate globally fitted K_D value measured.

2. The concentration range does not encompass a concentration that is at least 10-fold higher than the determined K_D .

We repeated the experiment with 5 sensorgrams using a concentration range 10-fold higher than the previous determined K_D (new Fig. 6k). In this new experiment B07-Fc K_D app = 78 nM.

b. Panel D and associated manuscript text

i. To support the conclusion that residues contained in region 30 to 41 of ACE2 of hamsters (especially Q34) are the molecular mechanism for a lower binding affinity of B07-Fc, they need to directly test the hypothesis. This can be done in a similar manner to what they have completed in panel A. One would expect the binding to be partially or fully restored by reverting the hamster residue to the human equivalent. Even better would be to also show that the inverse mutation leads to loss of activity. While the manuscript shows H35N, it is possible Q may be different. And could be tested. But to prove it is the dominant cause, the Q34 in hamster needs to be “reverted” to H34.

We previously made hACE2-H34Q mutation to test our hypothesis (old Supplementary Figure 3B). For clarity and to validate the importance of residue at position 34, we introduced a histidine at position 34 in hamster ACE2 (haACE2-Q34H). This mutant restored B07-Fc binding on haACE2. These data are now presented in Supplementary Fig. 9 and explained Lines 340-341.

ii. Also, the authors pass up a ripe opportunity to further map the epitope of B07 by taking advantage of the mouse ACE2. In panel B, they show that B07 binding to mouse ACE2 is just rising around $-4 \log \mu\text{g/mL}$ which is 1-2 logs different from the hamster and 3-4 from the human. This suggests that there are further differences between hamster and mouse in the epitope. Panel D, shows that mouse and hamster have the exact same residues as each other in aa 30-41, suggesting that differences exist in the epitope outside this region. The authors could perform the same assay as

in panel A but focused on aa residues differences found between hamster and mouse in the region analyzed for human allelic differences

We thank the reviewer for this comment. We made a mistake in typing the hamster ACE2 sequence. We have now corrected it in Supplementary Fig. 9b. The hamster and mouse ACE2 sequences between residues 27 and 42 are different at position 30-31. Residues NN30-31 in the mouse sequence likely explains the lower affinity of B07-Fc for mouse ACE2 compared to hamster ACE2. We explained this on Lines 343-346. To address this issue, we tested the impact of DK30-31NN substitution on B07-Fc binding by introducing NN residues at position 30-31 in human ACE2. The corresponding mutant is less recognized by B07-Fc compared to the WT hACE2 counterpart, confirming our assumption (Figure DK-NN below).

Figure DK-NN: VHH binding on cells expressing myc-hACE2 WT or myc-hACE2 DK30-31NN. Transfected cells were incubated with an anti-myc antibody (Abnova, 1/500) or B07-Fc (0.1 $\mu\text{g/ml}$) and stained with a AF647-conjugated anti-mouse antibody or a AF647-conjugated anti-human antibody respectively, before being analyzed by flow cytometry. Data are means \pm SD of three independent experiments.

33) Figure 7 and associated text

a. Please state in the method and legend if panels B and D are measuring genomic or subgenomic RNA.

Panels b and e of Fig. 7 are measuring genomic RNA. This is now specified in the Methods section (Line 872) and in the legend of Fig. 7b, e.

b. Please add the white light only images in a supplement for each of the fluorescence images panel C

We added the white light images for each fluorescent images of Fig. 7d in the Supplementary Fig. 10.

c. While differences in architecture of epithelium cells and bronchiole are interesting to see, it would be more informative to see immunohistochemical staining of the tissues for viral antigen compared to H&E sections of tissue. This more directly visual assess for the ability of the therapy to protect. Please provide this data

We agree that SARS-CoV-2 detection is a more direct way to observe the impact of B07-Fc on infection. We have now added immunostaining of the SARS-CoV-2 N nucleoprotein in Fig 7d (and supplementary Fig. 10). Viral N protein was detected in nasal respiratory epithelium, olfactory epithelium and lung bronchioli in VHH-ctl treated cells while it was undetectable in B07-Fc-treated cells 3 days pi. This suggested that VHH B07-Fc protects cells from infection, in addition to

protecting them from tissue damage. We mentioned this new result Line 358 and completed the Methods section (Line 840).

34) Supplemental figure 2. Add associated white light image for figure

We now added the white light images in the Supplementary Fig. 8c.

Answers to the reviewers

We thank the reviewers again for their feedback. Please find below (in blue) our responses.

REVIEWER COMMENTS

Reviewer #1 (Remarks to the Author):

This manuscript is strengthened with new data and structural insights, and the authors have address most of the reviewers' comments. Minor issues remain with grammar.

Minor comments:

1. Line 83, profile is misspelled as "profil".
2. Line 153, the correct expression should be "in cellulo" instead of "in cellula"

We have corrected the word "profile" in line 83, as suggested. However, we have retained the term "*in cellula*" in line 153, as it is the grammatically correct Latin meaning "in the cell" (from *cellula*, small chamber). While "*in cellulo*", is frequently encountered in the literature, it does not align, in our view, with proper Latin usage.

Reviewer #2 (Remarks to the Author):

The authors have thoroughly addressed the concerns raised in the initial review, and the revised manuscript is clear and scientifically sound. I recommend it for publication.

We thank the reviewer for this comment.

Reviewer #3 (Remarks to the Author):

Most of my concerns have been addressed, however there are still two that remain. I have listed them with the original comments, the author responses, and then my continued concern.

18. Line 167-169: There is quite a broad range of IC50's seen. Please discuss. Does this correlate with the affinity of ACE2 for the variant RBD's? Suggest adding a correlation analysis to the data set and add to manuscript figure or supplement.

Author Response:

As pointed out by the reviewer, we measured IC50s between 1937 nM (Delta) and 95 nM

(Omicron BA.2) for B07 and between 1180 nM (Delta) and 220 nM (BA.2) for B09. Several reports showed that the affinity of RBD for hACE2 tends to increase with new variants (for example, Pham Dan Lan, 2024; PMID: 38310477). The measured IC₅₀s of B07 for the different variants could be correlated with the affinity of hACE2 for RBDs. However, we did not find data comparing hACE2: RBDs' affinities for the different variants tested in our study. We consider this point too speculative to be mentioned.

Reviewer Response:

Because the VHH is kinetically competing for binding with the RBD for ACE2, a variant correlation analysis would improve the quality of the paper. This is because we would expect that a variant RBD with better affinity for ACE2 than the VHH would result in a worse variant-specific VHH IC₅₀ and vice versa. Readers of this paper will be familiar with the concept that ACE2 receptor affinity is changing with variants, as it has been discussed and evaluated many times in the literature. In addition to what was mentioned, a quick search came up with PMID:35093192, 35324257, 36399443. Thus, as a reader, I would not think this is a speculative question to be asking. It would be a question that I would expect to see analyzed.

Author Response:

The observed 1-log variations measured between IC₅₀ values (Fig. 4a) are likely due to the inherent limitations of dose-response assays and curve fitting, related to the relatively low affinity (about 200 nM) of the monomeric VHH for hACE2. These variations remain within an expected range and do not alter the conclusion that the VHH exhibit similar potency between the 7 variants tested. This broad-spectrum activity of VHHs is further validated by the results obtained with B07-Fc (Fig. 6g), in which we observed almost no variation in B07-Fc potency for the different variants. Of course, the way the spike engages ACE2 (affinity, positioning) could also account for the observed variations (see Figure 1 below). We have now clarified this point in the Discussion (Line 364-368). While it would be informative to link VHH potency to the affinity between the spike and hACE2 as suggested by the reviewer, such an analysis would require the following experiments, which are beyond the scope of our article:

- 1) to carry out experiments with new batches of VHH at much higher stock concentrations, which may increase the risk of VHH aggregation;
- 2) to systematically measure the spike/ACE2 binding affinity (K_D) in a single experiment, including the seven spike variants tested. Indeed, no published study has reported comparative K_D values for the variants we tested. Referring to a single study comparing the different variants is essential because K_D measurements are highly dependent on the methodology and the experimental conditions used (reported K_D values for a specific isolate vary from 3 nM to 237 nM, depending on the study, see PMID: 38310477). The references cited by the Reviewer (PMID:35093192, 35324257, 36399443) focus specifically on Delta and BA.1.

Figure 1: Footprints of B07 and RBDs on hACE2 for a selection of SARS-CoV-2 variants. The footprints of the VHH and RBD is represented as an outline colored in light blue (B07) and dark green (RBD) and superposed on the hACE2 structure showing their overlap (dashed area).

25. Figure 5 legend: Moi and hours of the assay readout are not the same between viruses for replication-based neutralization assays. This can cause the observed differences in IC50 values being reported between variants and B07-Fc. A higher moi means more virus is present and therefore more spike proteins. The consequence of this is that more spike proteins are present on particles that need to be neutralized and the result is a shift to a worse IC50 relative to a lower moi virus. Similarly, increase assay readout time means that additional replication cycles are occurring and this results in the generation of more virus genome copies. Since the assay is measuring viruses by RT-qPCR, it would mean that there could be a falsely worse IC50 value for B07-Fc for viruses assayed at 66 hours vs those assayed at 48 hours. The authors need to provide adequate justification of varying the assay moi and time differences and provide evidence that the differences do not affect IC50 values for each variant. In the absence of that, they will need to report the comparisons done under the same conditions to avoid biases.

Author Response:

MOI and incubation time were defined based on previous studies in order to obtain a similar rate of viral production after infection with the different isolates. We clarified this point in the Methods section (Line 758) (Touret, 2023 ; PMID: 36968074).

Reviewer Response:

Neutralization is a measure of the impact of an intervention on the virus entry. It is therefore highly impacted by changes in the number of virus particles present. This is because particle number is directly correlated with the number of spikes that are available to interact with ACE2. More spikes provide a higher probability that ACE2 will become bound and entry will occur when ACE2 is not being occupied by the VHH.

Using the rate of viral production after infection is not appropriate because it is a multifactorial output that is primarily reflective of the rate of viral replication and budding that occur after entry. For example, a poorly entering virus may compensate by having a higher rate of replication. Therefore, the justification to vary the multiplicity of infection (moi) to account for replication differences is not sufficient because it can lead to inaccurate conclusions due to confounding post-entry events.

If replication time is not a concern, then a reporter gene-based, immunofluorescence, or RT-PCR approach are all acceptable as long as the moi's are the same. However, based on the response,

it seems that replication times are a concern.

If the moi is kept constant and there remains a concern for differences in replication time, there are several approaches that can be taken. The first is to perform a plaque assay. With a plaque assay, slower replicating virus will have the same number of plaques as a faster one, they would just differ in the size of the plaque. A less technically challenging approach is to use single-cycle pseudovirus. These are well accepted in the field and used commonly to assay antibody neutralization. Lentiviral systems are most commonly used but there are other retroviral and VSV-based single-cycle systems.

Either way, modifying the moi and incubation time to account for replication differences is not an appropriate way to perform a neutralization assay. Therefore, the neutralization assays need to be redone as either pseudovirus entry assays or plaque assay.

Author Response:

We agree that the RT-qPCR based-assay as described in the manuscript (Supplementary Fig. 8e) does not represent a typical “neutralization assay” in the strict sense highlighted by the reviewer.

We performed this RT-qPCR viral inhibition assay to monitor the effect of B07-Fc on overall viral infection/multiplication, which includes the steps of (i) viral entry and fusion, (ii) viral multiplication and production after infection (transcriptional and translational levels), and (iii) virus budding.

Using this RT-qPCR viral inhibition assay, we validated the effect of B07-Fc on SARS-CoV-2 infection obtained by measuring syncytia formation by S-fuse (Fig. 6g). We further validated this effect by viral N protein detection assay after immunostaining in IGROV-1 cells (Fig. 6h).

As a reminder, the S-Fuse assay, developed by O. Schwartz’ lab, is a widely used assay to monitor syncytia formation after cell-cell fusion. This assay, based on the fusion between SARS-CoV-2-infected cells that express the spike protein at their surface and neighboring ACE2-positive cells, provides a quantitative assessment of viral infection (PMID: 33051876). Its reliability is endorsed in numerous published articles evaluating the impact of entry inhibitors (especially antibodies) (i.e. PMID: 33772244; PMID: 34237773; PMID: 34601723; PMID: 35290827; PMID: 35322239). We previously reported a correlation between neutralization titers obtained with the S-Fuse assay, a pseudovirus neutralization assay, and microneutralization assays (PMID: 33288662; PMID: 34282461).

We understand the concern that variability in incubation times and MOIs could potentially impact IC₅₀ values in RT-qPCR-based inhibition assays. To avoid potential ambiguity regarding the purpose of this assay (to validate B07-Fc inhibitory activity using a third replication/infection inhibition assay), and given that Supplementary Fig. 8e was based on one experiment only, we removed the IC₅₀ values from the figure (see new Supplementary Fig. 8e). Alternatively, and if preferred by the reviewer, we can also present only experiments performed using identical MOIs (0.07 PFU/Vero cells) and the same incubation time (48 hours) (see Figure 2 below). We also revised the Methods section to provide a clearer and more detailed explanation (Line 654-660). All IC₅₀ values were normalized using remdesivir as a positive control (defined as 100 % inhibition) and PBS as a negative control. This normalization further helps to account for any experimental variability across conditions.

Figure 2: Inhibition of SARS-CoV-2 replication by B07-Fc measured by RT-qPCR. Vero E6 cells pre-incubated with B07-Fc were infected with Delta B.1.617.2 or XBB.1.16.1 (MOI 0.07, 48 h) and viral replication was measured by quantitative RT-qPCR (one experiment in triplicates).

Finally, the reviewer suggested that it would also be of interest to use pseudovirus or plaque assays to compare IC₅₀ values and to overcome potential differences in viral replication. While each system has its strengths and limitations, we have previously shown that IC₅₀ values for entry inhibitors are consistent between pseudotyped particles and authentic SARS-CoV-2 (PMID: 32817357), as well as between RT-qPCR and plaque-based assays (PMID: 38883823).

To further support this point, we performed a simplified plaque assay using 2 variants (without including the VHH ctl) (see **Figure 3** below). We confirmed that IC₅₀ values measured for B07-Fc are consistent between S-Fuse assay (IC₅₀_{Delta} = 2.86 nM), RT-qPCR (IC₅₀_{Delta} = 10.23 nM), and plaque assays (IC₅₀_{Delta} = 3.87 nM). As expected, this fourth readout also confirmed the potency of B07-Fc on these 2 variants.

Figure 3: Inhibition of SARS-CoV-2 infection by B07-Fc measured by plaque assay. Vero E6 cells pre-incubated with B07-Fc were infected with Delta or KP.3.3 for 3 days. After fixation, plaques were visualized using crystal violet solution (one experiment in triplicates). To calculate the plaque forming units (PFU) per mL, counts from triplicate wells were averaged, and the average was multiplied by the dilution factor and the volume of inoculum plated.

In our view, the three independent readouts used in the manuscript (S-Fuse, N protein immunostaining, and RT-qPCR) provide a robust and convergent validation of the inhibitory effect of B07-Fc *in vitro*.